# Scalable Inference in SDEs by Direct Matching of the Fokker–Planck–Kolmogorov Equation

**Arno Solin**
Aalto University
Espoo, Finland
`arno.solin@aalto.fi`

**Ella Tamir**
Aalto University
Espoo, Finland
`ella.tamir@aalto.fi`

**Prakhar Verma**
Aalto University
Espoo, Finland
`prakhar.verma@aalto.fi`

## Abstract

Simulation-based techniques such as variants of stochastic Runge–Kutta are the *de facto* approach for inference with stochastic differential equations (SDEs) in machine learning. These methods are general-purpose and used with parametric and non-parametric models, and neural SDEs. Stochastic Runge–Kutta relies on the use of sampling schemes that can be inefficient in high dimensions. We address this issue by revisiting the classical SDE literature and derive direct approximations to the (typically intractable) Fokker–Planck–Kolmogorov equation by matching moments. We show how this workflow is fast, scales to high-dimensional latent spaces, and is applicable to scarce-data applications, where a non-parametric SDE with a driving Gaussian process velocity field specifies the model.

## 1 Introduction

Differential equations are the standard method of modelling *change* over time. In deterministic systems the dynamics specifying how the system evolves, are typically written in the form of an ordinary differential equation (ODE). The dynamics act as prior knowledge and often stem from first-principles in application areas such as physics, control engineering, chemistry, or compartmental models in epidemiology and pharmacokinetics. Recently, learning ODE dynamics with modern automatic differentiation packages in machine learning has awakened an interest in black-box learning of continuous-time dynamics (*e.g.*, [6, 37]) and enabled their more general use across time-series modelling applications.

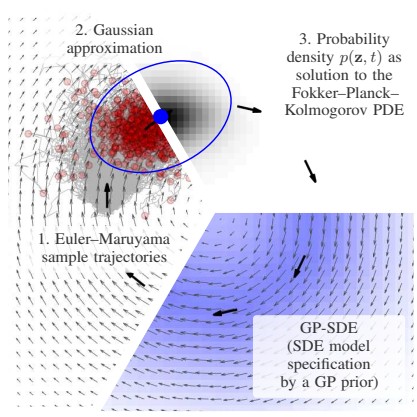

Figure 1: Views into solutions to SDEs.

A stochastic differential equation (SDE, [30, 40]) can be seen as a generalization of ODEs to stochastic dynamical settings, where the driving forces fluctuate or are uncertain. Stochastic dynamics appear naturally in applications where small (and typically unobserved) forces interact with the process, such as tracking applications, molecule motion, gene modelling, or stock markets. In machine learning, SDE models have received wide-spread attention due to their robustness and appealing properties for uncertainty quantification.

The concept of a 'solution' to an SDE is broader than that of an ODE. As the process is stochastic, the full solution entails a probability distribution, $p(\mathbf{z}, t)$, depending on time $t$ and covering the space $\mathbf{z}$ (see, *e.g.*, [36]). For Itô type SDEs, the evolution of the probability mass can be described in terms of the Fokker–Planck–Kolmogorov (FPK) partial differential equation (backward Kolmogorov equation). This equation is typically intractable, and instead the *de facto* approach for inference in SDEs in machine learning is sampling. The most common approaches in this space are based on

35th Conference on Neural Information Processing Systems (NeurIPS 2021).

Stochastic Runge–Kutta schemes (such as the *Euler–Maruyama scheme*) which are derived from the Itô–Taylor series. These schemes sample realization trajectories of the SDE by driving the dynamics with numerical simulation of Brownian motion. However, these schemes suffer from drawbacks both related to (ordinary) Runge–Kutta methods—such as step-size and sensitivity to stiffness—as well as problems associated with any sampling schemes, such as a high number of samples required for an accurate representation of the underlying distribution.

Despite these problems, few contemporary SDE approaches in machine learning explore SDE solutions beyond stochastic Runge–Kutta (or even the Euler–Maruyama scheme). Our aim is to try to broaden this view, and in Fig. 1 we sketch an example where we show three alternative solution perspectives to a Gaussian process prior SDE model (GP-SDE): the FPK probability density field, Euler–Maruyama samples, and a Gaussian assumed density approximation. We argue that a Gaussian approximation in latent space SDEs is reasonable, as Gaussian approximations are typically employed anyway in observation models, and allow for speeding-up learning by an order of magnitude.

The contributions of this paper are as follows. *(i)* We go through the workflow connecting 'random ODE' models with Itô SDEs driven by a Gaussian process prior over the velocity field, which allows for convenient specification of prior knowledge on the vector field and induces an implicit prior over the SDE trajectories; *(ii)* We revisit the classical SDE literature and derive direct approximations to the (typically intractable) Fokker–Planck–Kolmogorov equation in an assumed density Gaussian form that avoids sampling-based inference in the latent space, which makes inference fast and does not require sampling a high number of trajectories; *(iii)* We show how this workflow is fast, applicable to scarce-data applications, and how it also extends to previously presented latent SDE models.

## 1.1 Related Work

Neural ODEs [6] model ODE dynamics by a neural network. Such models were developed further in [37], where the encoder is an ODE-RNN that improves modelling of irregularly sampled time series. A latent Bayesian neural ODE model, ODE$^2$VAE, was examined in Yıldız et al. [49], where an encoder is combined with an ODE model whose second order dynamics are given by a Bayesian neural network. The neural ODE paradigm of modelling latent dynamics has been expanded to neural SDEs [26, 12, 46, 19], where the typical workflow is that a variational autoencoder (VAE, [21, 35]) is combined with a latent neural SDE, whose drift and diffusion are modelled by neural networks. In addition to modelling time series, neural SDEs have been used in generative models [16, 43, 44], where the generation of images from noise is modelled as the reverse-time process of a diffusion SDE by using Langevin dynamics on score-based models. Continuous normalizing flows are another model family, which applies ODE dynamics in a generative model [6, 11].

These works leverage simulation/sampling for solving the SDE in the latent space. The model can be trained by a stochastic adjoint method [26, 18]. More recently, latent neural SDEs have been trained deterministically by moment matching [27]. However, they discretized the system before matching the moments, while we form a direct approximation to the solution of the FPK. Compared to optimizing the moment ODEs, as discussed in [27], by maximizing likelihood, we regularize during inference by Gaussian process priors, or prior stochastic processes as in [26]. Approximative solutions to non-linear SDEs have been applied earlier in filtering theory, where the optimal filter is approximated by a Gaussian assumed density filter [23]. In [39], such approximations are used for continuous-discrete state-space modelling. An alternative to assumed density filters are local linearization methods [31, 41], and simulation-based Itô–Taylor series solutions, stochastic Runge–Kutta methods, and leapfrog methods such as Verlet for second-order SDEs (see [22, 40]). The approximations presented in this work are also related to GP approximations [4, 2] of SDEs. The linearization approximations are related to statistical linearization [10, 42], and variational approximations [4].

Orthogonally to the SDE inference, we also consider SDE model specification in terms of GP priors. The seminal work by Ruttor et al. [38] considered GP-SDE models with unit diffusion. Yildiz et al. [48] built a model, where the drift and diffusion are sparse Gaussian processes with time-independent kernels. In Hegde et al. [13], a spatio-temporal SDE with GP priors for the drift and diffusion was combined with a GP as a continuous version of deep Gaussian processes. State-space models with a GP latent state transition function [8] train a non-parametric latent process to approximate unobserved dynamics. These are related to hierarchical GP dynamics [45, 24], where prior knowledge of the system can be encoded in multi-level hierarchies, for modelling, *e.g.*, walking dynamics.

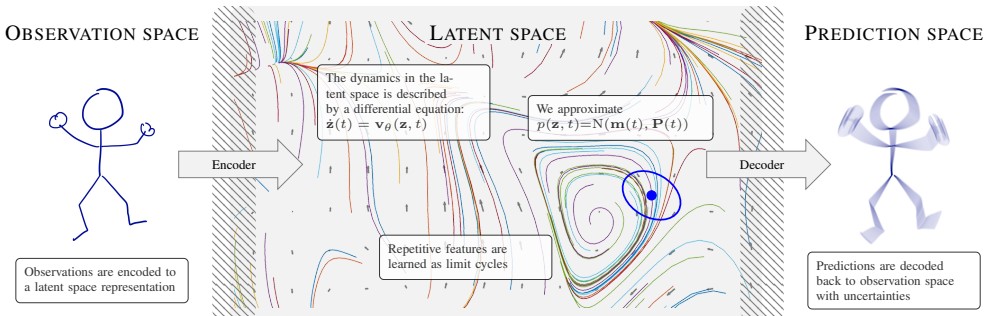

Figure 2: Latent dynamics workflow. The observations (left) are encoded into the latent space, where the dynamics of the system are learned as an SDE model. We approximate the solution to the SDE by a Gaussian for which we can approximate the dynamics of the first moments by an ODE system, thus avoiding sampling in the latent space. Predictions are finally mapped back.

## 2 Methods

We draft the methodology based on the latent dynamics workflow components as presented in Fig. 2. The focus is first on specifying models for the latent space dynamics, starting from implicit priors in terms of random ODEs which we frame as Itô SDEs. Thereafter, the focus shifts from model specification to inference, where we show that the Fokker–Planck–Kolmogorov equation can be efficiently approximated in an assumed density form, and finally brings us to cover the likelihood structure of these models as well.

### 2.1 Random Field Ordinary Differential Equations as SDEs

The continuous dynamics of a latent (unobserved) $\mathbf{z}(t) \in \mathbb{R}^d$ can conveniently be written in the form of a general first-order ordinary differential equation

$$\frac{\mathrm{d}}{\mathrm{d}t}\mathbf{z}(t) = \mathbf{v}_\theta(\mathbf{z}(t), t), \tag{1}$$

where $\mathbf{v}_\theta(\cdot) : \mathbb{R}^d \times \mathbb{R}_+ \to \mathbb{R}^d$ denotes the velocity field parametrized by $\theta$. This is a general form of a non-linear ODE system, where the dynamics are deterministic and fully characterized by $\mathbf{v}_\theta(\cdot)$. The methodology presented in this section directly extends to the case where the vector field $\mathbf{v}_\theta$ is time-dependent, but we omit the time dimension for simplicity of notation. Previously, the implicit prior on $\mathbf{z}(t)$ over $t$ specified by Eq. (1) has been generalized to stochastic models by considering $\mathbf{v}_\theta(\cdot)$ to be stochastic. These models are known as 'random' ODE models, and the random field $\mathbf{v}_\theta(\cdot)$ is typically either characterized by a Gaussian random field or Gaussian process model (see, *e.g.*, [38, 13]) or some parametric model (*e.g.*, [26]).

We consider an unconventional ODE model (or actually no ODE model at all, to be precise), where we specify a GP prior [34] over the velocity field in form of a multi-output Gaussian process prior:

$$\mathbf{v}(\mathbf{z}, t) \sim \mathrm{GP}(\boldsymbol{\mu}(\mathbf{z}), \boldsymbol{\kappa}(\mathbf{z}, \mathbf{z}')), \tag{2}$$

where $\boldsymbol{\mu} : \mathbb{R}^d \to \mathbb{R}^d$ is a mean function and $\boldsymbol{\kappa} : \mathbb{R}^d \times \mathbb{R}^d \to \mathbb{R}^{d \times d}$ is a matrix-valued covariance function. The Gaussian process prior is completely specified by its mean and covariance function, which encapsulate the assumptions about the sample processes/fields $\mathbf{v}$ (such as continuity, differentiability, curl, divergence, *etc.*): $\boldsymbol{\mu}(\mathbf{z}) := \mathrm{E}[\mathbf{v}(\mathbf{z})]$ and $\boldsymbol{\kappa}(\mathbf{z}, \mathbf{z}') := \mathrm{E}[(\mathbf{v}(\mathbf{z}) - \boldsymbol{\mu}(\mathbf{z}))(\mathbf{v}(\mathbf{z}') - \boldsymbol{\mu}(\mathbf{z}'))^*]$. In Fig. 3, we will consider examples of useful vector-valued covariance functions that encode properties on the vector field. For inference, the GP is conditioned on input–output pair observations $\mathcal{D} = \{(\mathbf{z}_i, \Delta\mathbf{z}_i)\}_{i=1}^n$ of the vector field, where $\Delta\mathbf{z}_i$ represents the observed derivative at $\mathbf{z}_i$. The conditioned vector field representation for an arbitrary point $\mathbf{z}_*$ in the latent space can be given by

$$\mathbf{v}(\mathbf{z}_*) \,|\, \mathcal{D} \sim \mathrm{GP}(\mathrm{E}[\mathbf{v}(\mathbf{z}) \,|\, \mathcal{D}], \mathrm{Cov}[\mathbf{v}(\mathbf{z}) \,|\, \mathcal{D}]), \tag{3}$$

where the $\mathrm{E}[\cdot]$ and $\mathrm{Cov}[\cdot]$ denote the marginal mean and (co)variance (for the multi-output GP, which means that the marginals are vector-valued). These take the form [34]: $\mathrm{E}[\mathbf{v}(\mathbf{z}_*)] = \mathbf{K}_* \hat{\mathbf{K}}^{-1} \mathbf{y}$ and

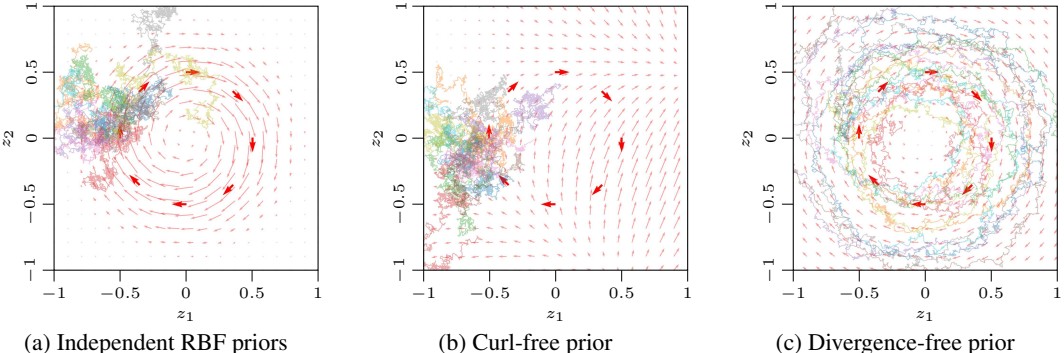

(a) Independent RBF priors      (b) Curl-free prior      (c) Divergence-free prior

Figure 3: Effect of different GP priors in an GP-SDE with 8 observations (large red arrows) to the GP posterior (small red arrows): (a) Shows results for independent RBF priors over $z_1$ and $z_2$; (b) shows results for the curl-free covariance function (encoding 'loop-aversion'); (c) shows results for the divergence-free covariance function (encoding 'energy preservation'). The hyperparameters ($\ell = 0.2, \sigma^2 = 0.1$) are the same in each.

$\mathrm{Cov}[\mathbf{v}(\mathbf{z}_*)] = \boldsymbol{\kappa}(\mathbf{z}_*, \mathbf{z}_*) - \mathbf{K}_* \hat{\mathbf{K}}^{-1} \mathbf{K}_*^\top$, where $\mathbf{y}$ are the stacked observations of the derivatives such that $\mathbf{y} = (\Delta \mathbf{z}_1^\top, \ldots, \Delta \mathbf{z}_n^\top)^\top$. The Gram matrix $\mathbf{K}$ corresponds to evaluations of the covariance function such that $\mathbf{K}_{ij} = \boldsymbol{\kappa}(\mathbf{z}_i, \mathbf{z}_j)$ are sub-blocks of $\mathbf{K}$ corresponding to the observation pairs $(i, j)$, and $\hat{\mathbf{K}} = \mathbf{K} + \gamma \mathbf{I}$, where $\gamma$ is a nugget (observation noise/discrepancy) term, and $\mathbf{K}_*$ is the cross-covariance between $z$ and $z_*$. The prohibitive cubic computational scaling associated with GP models manifests in the inversion of $\hat{\mathbf{K}}$, and thus for large $n$, approximations based on inducing points or projections are used in practice to avoid this explicit inversion. In the light of Eq. (1) with $\mathbf{v}(\mathbf{z}) \sim \mathrm{GP}(\cdot, \cdot)$, the ODE is driven by a multi-dimensional Gaussian random field conditioned on $\mathcal{D}$. A straightforward way of dealing with a model of this kind, is to do inference by sampling random draws of the velocity field from the GP, and then drive the ODE with those samples (can be viewed as an Monte Carlo approach for drawing ODE realizations).

However, a more convenient way is to specify the prior over the stochastic dynamics in a stochastic differential equation form. At its core, a lot of previous work in this space hinges on the realization that if everything is essentially Gaussian, an equivalent model can be specified in terms of an Itô SDE describing the stochastic evolution of trajectories affected by the GP velocity field [see 13, for discussion]. Informally, this takes the white noise form $\frac{\mathrm{d}}{\mathrm{d}t}\mathbf{z}(t) = \mathbf{f}(\mathbf{z}) + \mathbf{L}(\mathbf{z})\,\mathbf{w}(t)$, where $\mathbf{f}(\mathbf{z}) = \mathrm{E}[\mathbf{v}(\mathbf{z})]$ and $\mathbf{L}(\mathbf{z})$ denotes a square-root factor such that $\mathbf{L}\mathbf{L}^\top = \mathrm{Cov}[\mathbf{v}(\mathbf{z})]$ (in the scalar case, just the square-root, and in the multi-output case, $e.g.$, the Cholesky factor). Here $\mathbf{w}(t)$ is a white noise process with unit spectral density. It is worth noting that we do not give guarantees for a direct link between the random ODE in Eq. (2) and the following SDE formulation (see App. A.3 for discussion). Yet, formally we write a similarly-behaving SDE in the standard Itô SDE form:

$$\mathrm{d}\mathbf{z}(t) = \mathbf{f}(\mathbf{z}, t)\,\mathrm{d}t + \mathbf{L}(\mathbf{z}, t)\,\mathrm{d}\boldsymbol{\beta}(t), \tag{4}$$

where $\mathrm{d}\boldsymbol{\beta}(t)$ is vector-valued unit Brownian motion (the spectral density $\mathbf{Q}$ is set to $\mathbf{I}$). For a GP-SDE, the drift is driven by the GP mean, $\mathbf{f}(\mathbf{z}, t) := \mathrm{E}[\mathbf{v}(\mathbf{z})]$ and the diffusion by the square-root factor of the marginal covariance at $\mathbf{z}(t)$, $\mathbf{L}(\mathbf{z}, t) := \sqrt{\mathrm{Cov}[\mathbf{v}(\mathbf{z})]}$. To be precise, the GP-SDE drift and diffusion at a point $\mathbf{z}_*$ are determined by the GP predicted mean and variance at $\mathbf{z}_*$, which can be written as

$$\mathbf{f}(\mathbf{z}_*, t) = \mathrm{E}[\mathbf{v}(\mathbf{z}_*)] = \mathbf{K}_* \hat{\mathbf{K}}^{-1} \mathbf{y} \;\; \text{and} \;\; \mathbf{L}(\mathbf{z}_*, t) = \sqrt{\mathrm{Cov}[\mathbf{v}(\mathbf{z}_*)]} = \sqrt{\boldsymbol{\kappa}(\mathbf{z}_*, \mathbf{z}_*) - \mathbf{K}_* \hat{\mathbf{K}}^{-1} \mathbf{K}_*^\top}. \tag{5}$$

## 2.2 Fokker–Planck–Kolmogorov Equation

We are interested in *solving* SDE models of the form in Eq. (4), but without the restriction that the drift and diffusion are defined by a Gaussian process, and present the related theory with a model-agnostic view on the problem. Because the resulting solutions are stochastic processes, the full solution to the SDE can be characterized by its time-evolving probability density function. Let $\mathbf{z}(t_0) \sim p(\mathbf{z}(t_0))$ be some initial condition which we assumed to be independent of the Brownian motion. The probability

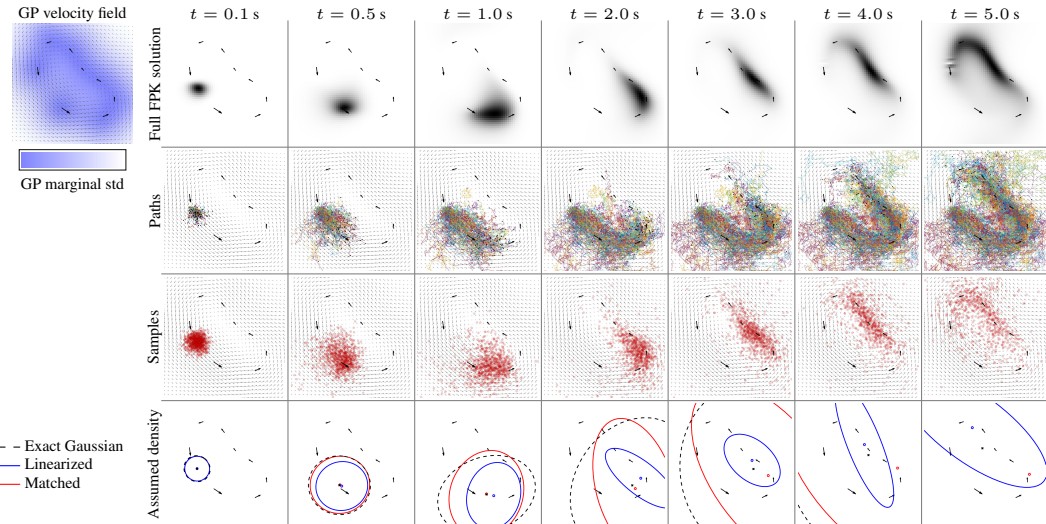

Figure 4: Approximations to the FPK equation: The **top-left** figure shows the 8 observations (black arrows) and the inferred GP velocity (grey arrows, marginal uncertainty in shaded blue). The **top-row** shows the progression of the probability mass $p(\mathbf{z}, t)$ following the Fokker–Planck–Kolmogorov equation. The **middle rows** show Euler–Maruyama sample trajectories for the problem, and the **bottom row** compares the two assumed density approximations to the exact Gaussian approximation of the FPK solution. The bottom row ellipses are $95\%$ confidence regions.

density $p(\mathbf{z}, t)$ of the solution of the SDE in Eq. (4) solves the Fokker–Planck–Kolmogorov (FPK) partial differential equation (PDE):

$$\frac{\partial p(\mathbf{z}, t)}{\partial t} = -\sum_i \frac{\partial}{\partial z_i} [f_i(\mathbf{z}, t)\, p(\mathbf{z}, t)] + \frac{1}{2} \sum_{i,j} \frac{\partial^2}{\partial z_i\, \partial z_j} \left\{ [\mathbf{L}(\mathbf{z}, t)\, \mathbf{Q}\, \mathbf{L}^\top(\mathbf{z}, t)]_{ij}\, p(\mathbf{z}, t) \right\}. \quad (6)$$

For a proof, see [40]. This PDE is also known as the Fokker–Planck equation (in physics) and the forward Kolmogorov equation (in stochastics). An appealing alternative form [40, Sec. 5.3] of the FPK equation can be given in terms of the following evolution equation with the adjoint operator $\mathcal{A}^*$:

$$\frac{\partial p}{\partial t} = \mathcal{A}^* p, \text{ with } \mathcal{A}^*(\bullet) = -\sum_i \frac{\partial}{\partial z_i} [f_i(\mathbf{z}, t)\, (\bullet)] + \frac{1}{2} \sum_{i,j} \frac{\partial^2}{\partial z_i\, \partial z_j} \{ [\mathbf{L}(\mathbf{z}, t)\, \mathbf{Q}\, \mathbf{L}^\top(\mathbf{z}, t)]_{ij}\, (\bullet) \}.$$
$$(7)$$

Eq. (7) allows for various kind of approaches for direct approximation of the FPK equation either by basis function approximations, finite differences, or other methods (Sec. 9.6 in [40] provides examples of using point collocation, Ritz–Galerkin, and FEM type of methods for approximating the solution). For example, the results in Fig. 1 and Fig. 4 are estimated by a grid discretization over $\mathbf{z}$ and solving the resulting (finite-dimensional) ODE corresponding to Eq. (7) by the matrix exponential: $\mathbf{p}(t) = \exp(\mathbf{A}(t - t_0))$. See App. A.2 for details. Even if the widely-used Euler–Maruyama, Milstein, and more general stochastic Runge–Kutta schemes are derived from the Itô–Taylor series, the resulting methods can still be viewed as an approximation of $p(\mathbf{z}, t)$.

## 2.3 Assumed Density Approximation of FPK

For the purpose of modelling latent space dynamics of systems of the kind in Fig. 2, we note that there the common practice of solving the latent SDE through costly simulation/sampling and then employing a variational (Gaussian) approximation in the encoder/decoder seems contradictory. That is, it might be unnecessary to sample realizations of the trajectory dynamics, if the interest is only in the time-marginals of the process. Thus, we seek to directly characterize the first two moments of the solution to the FPK equation in Sec. 2.2. We replace the FPK solution with a Gaussian approximation of form

$$p(\mathbf{z}, t) \approx \mathrm{N}(\mathbf{z} \mid \mathbf{m}(t), \mathbf{P}(t)), \quad (8)$$

where $\mathbf{m}(t)$ and $\mathbf{P}(t)$ are interpreted as a mean and covariance of the state of the solution at time $t$. This kind of approximation is commonly referred to as a Gaussian *assumed density* approximation (see, *e.g.*, [23, 39]), because the computations are done under the assumption that the state distribution is Gaussian. Assumed density approximations are common in signal processing–driven SDE methodology, and we refer the reader to Sec. 9.1 in [40] for a detailed overview. Following [39], we revisit the idea that a Gaussian process approximation to the SDE Eq. (4) can be obtained by integrating the following differential equations from the initial conditions $\mathbf{m}(t_0) = \mathrm{E}[\mathbf{z}(t_0)]$ and $\mathbf{P}(t_0) = \mathrm{Cov}[\mathbf{z}(t_0)]$ to the target time $t$:

$$\frac{\mathrm{d}\mathbf{m}}{\mathrm{d}t} = \int \mathbf{f}(\mathbf{z}, t)\, \mathrm{N}(\mathbf{z}\,|\,\mathbf{m}, \mathbf{P})\, \mathrm{d}\mathbf{z} \quad \text{and} \tag{9}$$

$$\frac{\mathrm{d}\mathbf{P}}{\mathrm{d}t} = \int \mathbf{f}(\mathbf{z}, t)\, (\mathbf{z} - \mathbf{m})^\top\, \mathrm{N}(\mathbf{z}\,|\,\mathbf{m}, \mathbf{P})\, \mathrm{d}\mathbf{z}$$
$$+ \int (\mathbf{z} - \mathbf{m})\, \mathbf{f}^\top(\mathbf{z}, t)\, \mathrm{N}(\mathbf{z}\,|\,\mathbf{m}, \mathbf{P})\, \mathrm{d}\mathbf{z} + \int \mathbf{L}(\mathbf{z}, t)\, \mathbf{Q}\, \mathbf{L}^\top(\mathbf{z}, t)\, \mathrm{N}(\mathbf{z}\,|\,\mathbf{m}, \mathbf{P})\, \mathrm{d}\mathbf{z}. \tag{10}$$

These equations for the evolution of the first moments of the solution to the SDE can be interpreted as expectations over the drift and diffusion dynamics of the SDE, and can be derived from the FPK in Eq. (6). Conveniently, these expressions are *not* stochastic, but instead take the form of an ODE system that—given the integrals are tractable—can be solved with out-of-the-box ODE solvers. However, even if Eqs. (9) and (10) provide a generic Gaussian assumed density approximation framework for SDEs, an implementation of the method requires solving the following kind of $d$-dimensional Gaussian integrals:

$$\mathrm{E}_\mathrm{N}[\bullet] = \int [\bullet]\, \mathrm{N}(\mathbf{z}\,|\,\mathbf{m}, \mathbf{P})\, \mathrm{d}\mathbf{z}. \tag{11}$$

In the following sections we will consider two approaches (local linearization and moment matching with symmetric quadrature) which scale linearly in the number of latent dimensions $d$.

## 2.4 Linearizing the FPK Equation

Local linearization around the $\mathbf{m}$ (via a Taylor series approximation) is a classical approach widely used for this type of Gaussian integrals in machine learning and filtering theory [17, 29]. If the function $\mathbf{f}(\mathbf{z}, t)$ is differentiable, the covariance differential equation can be simplified by using Stein's lemma [32] such that

$$\int \mathbf{f}(\mathbf{z}, t)\, (\mathbf{z} - \mathbf{m})^\top\, \mathrm{N}(\mathbf{z}\,|\,\mathbf{m}, \mathbf{P})\, \mathrm{d}\mathbf{z} = \left[\int \mathbf{F}_\mathbf{z}(\mathbf{z}, t)\, \mathrm{N}(\mathbf{z}\,|\,\mathbf{m}, \mathbf{P})\, \mathrm{d}\mathbf{z}\right] \mathbf{P}, \tag{12}$$

where $\mathbf{F}_\mathbf{z}(\mathbf{z}, t)$ is the Jacobian of $\mathbf{f}(\mathbf{z}, t)$ with respect to $\mathbf{z}$. Linearizing around the mean $\mathbf{m}$ and approximating the diffusion as $\mathbf{L}(\mathbf{z}, t) \approx \mathbf{L}(\mathbf{m}, t)$ gives a linearized form of Eqs. (9) and (10):

$$\frac{\mathrm{d}\mathbf{m}}{\mathrm{d}t} = \mathbf{f}(\mathbf{m}, t) \quad \text{and} \quad \frac{\mathrm{d}\mathbf{P}}{\mathrm{d}t} = \mathbf{P}\, \mathbf{F}_\mathbf{z}^\top(\mathbf{m}, t) + \mathbf{F}_\mathbf{z}(\mathbf{m}, t)\, \mathbf{P} + \mathbf{L}(\mathbf{m}, t)\, \mathbf{Q}\, \mathbf{L}^\top(\mathbf{m}, t), \tag{13}$$

which provides a direct way of propagating the moments of the latent SDE through an ODE for the mean and covariance without the need of drawing multiple sample trajectories. The resulting ODE is $(d + d^2)$-dimensional, and only requires one evaluation of the drift, diffusion, and Jacobian per step.

## 2.5 Matching Moments of the FPK Equation

The local linearization approach given in the preceding section is efficient, but fully *local*. An alternative way of constructing an assumed density approximation to $p(\mathbf{z}, t)$ is to directly match the moments by solving the Gaussian integrals in Eqs. (9) and (10) by Gaussian quadrature methods. The approximation to Eq. (11) would take the form $\int \mathbf{g}(\mathbf{z}, t)\, \mathrm{N}(\mathbf{z}\,|\,\mathbf{m}, \mathbf{P})\, \mathrm{d}\mathbf{z} \approx \sum_i w^{(i)}\, \mathbf{g}(\mathbf{z}^{(i)}, t)$, for an arbitrary integrand $\mathbf{g}(\mathbf{z}, t)$, weights $w^{(i)}$, and so called sigma points $\mathbf{z}^{(i)} = \mathbf{m} + \sqrt{\mathbf{P}}\, \boldsymbol{\xi}_i$. Here $\sqrt{\mathbf{P}}$ denotes a square-root factor of $\mathbf{P}$ such as the Cholesky decomposition. The multi-dimensional Gaussian quadrature (or *cubature*, see [7]) rule is characterized by the evaluation points and their associated weights $\{(\boldsymbol{\xi}_i, w_i)\}$. We write Eqs. (9) and (10) in a Gaussian assumed density form which matches the moments by quadrature as follows [40]:

$$\frac{\mathrm{d}\mathbf{m}}{\mathrm{d}t} = \sum_i w^{(i)}\, \mathbf{f}(\mathbf{m} + \sqrt{\mathbf{P}}\, \boldsymbol{\xi}_i, t) \quad \text{and} \tag{14}$$

$$\frac{\mathrm{d}\mathbf{P}}{\mathrm{d}t} = \sum_i w^{(i)}\, \mathbf{f}(\mathbf{m} + \sqrt{\mathbf{P}}\, \boldsymbol{\xi}_i, t)\, \boldsymbol{\xi}_i^\top\, \sqrt{\mathbf{P}}^\top$$
$$+ \sum_i w^{(i)}\, \sqrt{\mathbf{P}}\, \boldsymbol{\xi}_i\, \mathbf{f}^\top(\mathbf{m} + \sqrt{\mathbf{P}}\, \boldsymbol{\xi}_i, t) + \sum_i w^{(i)}\, \mathbf{L}(\mathbf{m} + \sqrt{\mathbf{P}}\, \boldsymbol{\xi}_i, t)\, \mathbf{Q}\, \mathbf{L}^\top(\mathbf{m} + \sqrt{\mathbf{P}}\, \boldsymbol{\xi}_i, t). \tag{15}$$

The computational complexity of this approach is highly dependent on the choice of quadrature method. A typical choice in ML applications would be Gauss–Hermite quadrature, which factorizes over the input dimensions leading to an exponential number ($p^d$) of function evaluations/sigma points in the input dimensionality $d$ for a desired order $p$. In order to guarantee scalability, we employ a symmetric $3^{\text{rd}}$ order cubature rule [3] which similarly to Gauss–Hermite ($p = 3$) is exact for polynomials up to degree 3. The points are given by scaled unit coordinate vectors $\mathbf{e}_i$ such that

$$\boldsymbol{\xi}_i = \begin{cases} \sqrt{d}\,\mathbf{e}_i, & \text{for } i = 1, \ldots, d, \\ -\sqrt{d}\,\mathbf{e}_i, & \text{for } i = d+1, \ldots, 2d, \end{cases} \tag{16}$$

and the associated weights are $w_i = \frac{1}{2d}$. This approach provides a direct way of propagating the 'true' moments of the latent SDE through an ODE for the mean and covariance and without the need of drawing multiple sample trajectories. The resulting ODE is $(d + d^2)$-dimensional, and requires only $2d$ evaluations of the drift and diffusion per step.

## 2.6  Analysis of the Computational Complexity

In terms of the asymptotic computational complexity, the linearization approach in Sec. 2.4 requires $\mathcal{O}(1)$ evaluations of the drift, diffusion, and Jacobian per step. The moment matching approach in Sec. 2.5 requires $\mathcal{O}(d)$ evaluations of the drift and diffusion as well as an $\mathcal{O}(d^3)$ Cholesky decomposition per step. The simplest Monte Carlo simulation method with $p$ samples requires $\mathcal{O}(p)$ evaluations of the drift and diffusion per step. Additionally, the naïve requirement for $p$ grows exponentially in $d$. On the other hand, the simulation approach is fully parallelizable over $p$, while the moment matching approach the number of nonparallelizable operations is $\mathcal{O}(d^2)$ with the Cholesky decomposition being the bottleneck, and the linearization approach is nonparallelizable. While the linearization approach has the lowest number of function evaluations with respect to $d$, the cost of computing the Jacobian can be prohibitively large for arbitrarily complex models. Nevertheless, the Jacobian is available in closed-form for GPs and may be evaluated reasonably fast for neural network based drifts, see App. B.2 for empirical computational costs of evaluating the Jacobian when growing the network size. Thus we expect the FPK approximation schemes to always be beneficial in CPU-only cases (incl. CPU multi-threading and embedded devices). In multicore GPU use, for low-dimensional $d$, sampling remains appealing if GPU memory does not become a bottleneck.

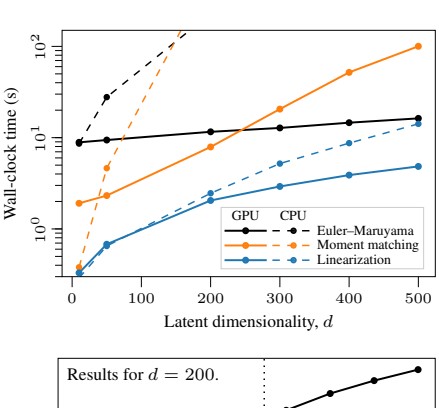

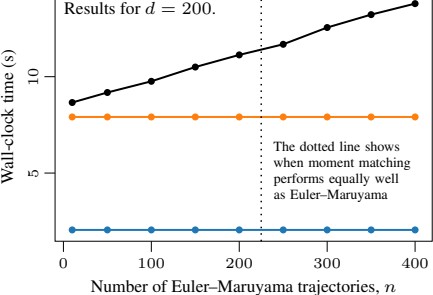

Figure 5: Empirical timing experiments with error of final margins matched.

## 3   Experiments

The goals of the experiments are three-fold: We first provide a study of the computational complexity. Then, we look into properties of the GP-SDE model from Sec. 2.1, where the experiments are concerned with showcasing model specification rather than inference. Finally, we consider two benchmark problems with high-dimensional inputs for learning a latent SDE model, where we test the performance of the approximations presented when the model is not defined by GPs, as the SDE methods presented in Sec. 2 are model-agnostic.

**Timing Experiments**   To confirm the analysis in Sec. 2.6 and provide a practical insight, we run numerical experiments with the error of final marginal mean/covariance controlled. We use a high-dimensional model of $d$ independent Beneš SDEs ($\mathrm{d}z(t) = \tanh(z)\,\mathrm{d}t + \mathrm{d}\beta(t)$, see [40]) with different $z_0$ per dimension. The model is non-linear and solution-space multi-modal, but both $p(\mathbf{z}, t)$ and the marginal moments $(\mathbf{m}(t), \mathbf{P}(t))$ are available in closed form (see App. B.1). In comparison

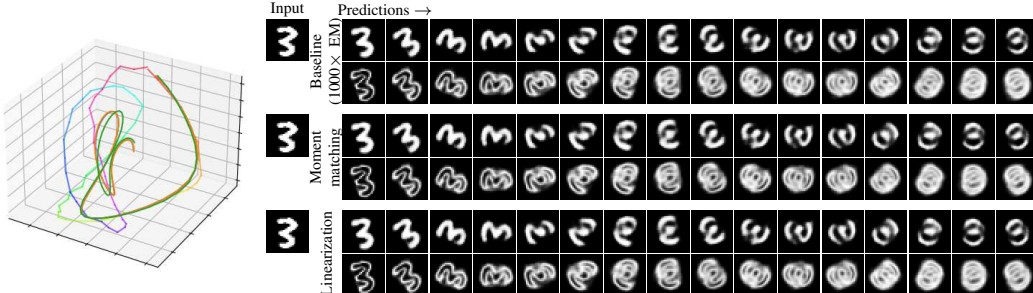

(a) Latent space trajectories                (b) Forward prediction

Figure 6: Results on rotating MNIST. (a) shows the latent space prediction mean trajectories for one test image. Evolution of the true trajectory is shown in HSV colour and the predicted trajectory by Euler–Maruyama, moment matching, and linearization scheme is shown in green, blue, and orange (all overlapping one another). (b) shows the progression of the prediction (mean and std dev) of the test image, when it traverses the learned dynamics in the latent space. Both the moment matching and linearization schemes match the baseline (with 1000 Euler–Maruyama trajectories).

to Euler–Maruyama, we control the number of trajectories $n$ by bounding the error (in terms of KL divergence) between Euler–Maruyama and the ground-truth to match the error in our moment matching approach, and consider the methods equivalent in terms of the quality of the solution. See App. B.1 for the required number of trajectories to match the KL divergence for each dimensionality plotted. Fig. 5 shows GPU/CPU wall-clock times (GPU: NVIDIA Tesla V100 32 GB with Intel Xeon Gold 6134 3.2 GHz; CPU: Xeon Gold 6248 2.50GHz). We implement the models in PyTorch [33] and report means of 10 repetitions (std negligible). In low-dimensional cases, both approximation methods outperform sampling, whereas in high dimensions GPU parallelisation becomes dominant and only the linearized approximation remains highly competitive. This example should favour sampling: In the Beneš model, the number of trajectories, $n$, per $d$ in Euler–Maruyama remains low ($n$ is linear in $d$), which is due to the diagonal (independent) diffusion matrix. In a correlated latent space $n$ would grow super-linearly (even exponentially), which would further push the difference between methods.

**GP-SDE Model Specification** We consider an GP-SDE model with just 8 observations of the dynamics, where the lack of data can be compensated for with encoding prior knowledge into the model. We use the model formulation given in Sec. 2.1, and study the effect of GP priors, the first of which is an independent squared exponential (RBF) prior for each dimension which encode continuity and smoothness in the velocity field. The second GP prior is the multi-dimensional curl-free kernel [47] (see App. B.3) which encodes the assumption of a curl-free random vector field. This property can be interpreted as 'loop aversion' in the GP-SDE context. The third prior, is a multi-dimensional divergence-free kernel [47] which encodes the assumption of no divergence in the random vector field. This property can be interpreted as 'energy preservation' or source-freeness. These properties are visible in Fig. 3, where the hyperparameters are fixed to same values for all models.

**Assumed Density Approximation of the FPK** We provide an illustrative example of the moment evolution methods in a GP-SDE model with 8 observations along a bean curve and independent squared exponential GP priors per dimension. As a baseline, we solve the FPK in Eq. (7) by finite-differences discretization in $\mathbf{z}$ (see App. A.2 for details). Fig. 4 shows the evolution of the SDE solution over the time-course of 5 s. The probability mass dissolves quickly, which is hard to interpret from the top-row figure alone. Comparison between the point clouds and the top row shows that even with 1000 trajectories and just a two-dimensional space, it is hard to capture detailed structure in the SDE solution. The bottom row compares the local linearization (Sec. 2.4) and the moment matching (Sec. 2.5) assumed density approximations to the exact Gaussian approximation of the FPK solution. The linearized approach is mode-seeking (matches local curvature), while the moment matching approach captures the overall structure of the optimal Gaussian approximation.

**Rotating MNIST** In the spirit of Fig. 2, we run the proposed methods on Rotating MNIST ([25], available under CC BY-SA 3.0), similar to [49, 5]. The data set consists of various handwritten digit '3's rotated uniformly in 64 angles. We train a VAE [21] first by freezing the latent space dynamics, allowing us to generate the latent samples for learning the dynamics by applying the

Table 1: Rotating MNIST results.

| INFERENCE SCHEME | MSE | NLPD ($t = 64$) |
|---|---|---|
| Euler–Maruyama | $0.046 \pm 0.006$ | $33.0 \pm 5.4$ |
| Moment matching | $0.051 \pm 0.007$ | $52.7 \pm 9.5$ |
| Linearization | $0.052 \pm 0.007$ | $54.5 \pm 9.9$ |

Table 2: Wall-clock timings for MOCAP.

| TIME/s±STD | | NUMBER OF E-M PATHS | | |
|---|---|---|---|---|
| LIN. | MOM. MAT. | 1 | 25 | 200 |
| GPU $2.1\pm.1$ | $6.0\pm.1$ | $37.1\pm.1$ | $39.9\pm.1$ | $40.4\pm.7$ |
| CPU $1.8\pm.1$ | $1.8\pm.1$ | $27.7\pm.7$ | $38.6\pm1.5$ | $94.2\pm3.5$ |

Table 3: Test MSE on 297 future MOCAP points averaged over 50 samples. 95% confidence interval reported based on t-statistic. [†]results from [49], [‡]results from [26]

| METHOD | TEST MSE |
|---|---|
| DTSBN-S [9] | $34.86 \pm 0.02$[†] |
| npODE [14] | $22.96$[†] |
| NeuralODE [6] | $22.49 \pm 0.88$[†] |
| ODE$^2$VAE [49] | $10.06 \pm 1.4$[†] |
| ODE$^2$VAE-KL [49] | $8.09 \pm 1.95$[†] |
| Latent ODE [37] | $5.98 \pm 0.28$[‡] |
| Latent SDE [26] | $4.03 \pm 0.20$[‡] |
| Latent SDE (assumed density) | $7.55 \pm 0.05$ |

trained VAE. Then freezing the VAE encoder/decoder and training a 16-dimensional GP-SDE model in the latent space with independent squared exponential GP priors (see App. B.5). In Fig. 6 we feed in one observation and let it follow the learned dynamics of rotation. As the baseline, we use 1000 trajectories computed using Euler–Maruyama. Fig. 6a demonstrates the model's capability to learn the latent trajectory, and we show the trajectories for all the methods in three latent dimensions with most variation. The trajectories for the three methods overlap, and qualitatively the results are identical in Fig. 6b. Quantitatively the brute-force sampling baseline gives slightly better MSEs over images and final-step mean negative log predictive densities (NLPD, see Table 1).

**Motion Capture Data** The CMU walking data set ([1], CMU MoCap available under CC BY-ND 4.0) is a real-world noisy data set with 50 sensors that track a human subject's walking. As in Yıldız et al. [49] and Li et al. [26], we model the sequences of a single subject, 35, for which there are 16 train set, three validation set and four test set sequences. The task is to predict the state of the system in the future given three initial points. In this experiment, we demonstrate that replacing SDE solver–based methods by an assumed density approximation, a latent neural SDE system can be learned efficiently without sampling trajectories. For this purpose, and for better comparability to earlier work, the latent SDE drift and diffusion are neural networks. As in Li et al. [26], we regularize the learned posterior process by a prior process. The loss function consists of three terms: reconstruction loss, VAE encoded initial position KL-divergence, and the KL-divergence between posterior/prior processes. The moments of the posterior SDE approximation are denoted by $\mathbf{m}(t), \mathbf{P}(t)$, those of the prior process $\mathbf{m}_*(t), \mathbf{P}_*(t)$, and the observation times by $\{t_j\}_{j=0}^m$. The loss becomes

$$\mathcal{L} = -\sum_{j=0}^m \log p(\mathbf{x}(t_j) \mid \mathbf{z}(t_j)) + \mathrm{D}_{\mathrm{KL}}\left[q(\mathbf{z}(t_0) \mid \mathbf{x}(t_0)) \,\|\, p(\mathbf{z})\right]$$
$$+ \sum_{j=1}^m \gamma\, \mathrm{D}_{\mathrm{KL}}\left[\mathrm{N}(\mathbf{m}(t_j), \mathbf{P}(t_j)) \,\|\, \mathrm{N}(\mathbf{m}_*(t_j), \mathbf{P}_*(t_j))\right], \quad (17)$$

where $\mathbf{z}(t_j)$ and $\mathbf{x}(t_j)$ are the latent codes and observations, respectively, $q$ denotes the conditional encoder distribution, $p(z)$ is the normal distribution, and $\log p(\mathbf{x}(t_j) \mid \mathbf{z}(t_j))$ is the model likelihood of the observations, given latent codes. While training, the parameters to optimize include those of a latent neural network which defines the prior and posterior dynamics, and of the VAE which encodes an initial point and decodes at each discrete time step corresponding to a train set frame. As the VAE is trained simultaneously to the latent dynamics, the approach does not provide a set of true latent samples to compare to, in contrast to the MNIST experiments. The VAE encoder prior is the normal distribution, whereas the encoder posterior is acting as the initial distribution for the latent SDEs. For the encoder design, we use a fully-connected neural network, which encodes the three first data points to create latent state and context vectors. As the prior process was a SDE with zero drift and $\sigma\mathbf{I}$ diffusion. The context is passed through the dynamics, similar to the treatment of velocity in Yıldız et al. [49] (see App. B.6).

The result in Table 3 is competitive considering that solving the latent SDE with the linearization approach roughly matches the required computation budget for *one* stochastic Runge–Kutta sample in the other methods. This is highlighted in Table 2 which shows the wall-clock times for the model used in this experiment both in a CPU and a GPU setting (see hardware description in previous experiment).

## 4   Discussion and Conclusions

In this paper our interest has been in both SDE model specification and approximative inference. We considered GP-SDE models for data-scarce applications that need injection of prior knowledge such as in Fig. 3. For inference, we built upon the established methodology of assumed density approximations in signal processing for directly approximating the solution distribution of an Itô SDE, where the methods apply to any latent space SDE models.

We put interest in *weak* solution concepts for SDEs. We considered both linearization and moment matching based methods for capturing the first two moments of the SDE solutions, which respectively require only $\mathcal{O}(1)$ and $\mathcal{O}(d)$ evaluations of the model functions per solver step in latent space dimensionality $d$. Furthermore, they only require solving of one ODE rather than simulating multiple trajectories. This makes them orders of magnitudes lighter than the current state-of-the-art in neural SDEs, which we analyzed both through theoretical bounds in Sec. 2.6, ran numerical experiments with the final error controlled (Fig. 5), and highlighted the practical wall-clock time in a MOCAP experiment. We further argued that a Gaussian assumption makes sense in applications of the form in Fig. 2 as one is typically employed anyway in the encoder–decoder.

There are some key differences between our assumed density approach and commonly used sampling approaches to solving neural SDE models. While stochastic Runge–Kutta methods are typically either concerned with strong (pathwise) or weak (in distribution) solutions, we leverage the even weaker solution concept of only tracking the first two moments of the solution. This simplification of turning solving the SDE into a deterministic ODE problem of its moments, comes with some remarkable computational savings, and often suffice in practical modelling. Then again, the proposed method does not lend itself well to cases where pathwise sample trajectories are required.

We recognize that we cannot give guarantees for the approximated Gaussian integrals to capture the evolution of the true moments outside particular special cases (*e.g.*, 3rd order cubature is exact for polynomials up to order 3). Yet the performance on practical applications is considered reliable, and these kinds of approached are commonly employed across assumed density filtering in signal processing. The use of the approximations schemes we present inherently makes the assumption that the time-marginals of the process are of higher interest compared to the pathwise solutions to the SDE. While the sampling-based methods are less efficient than assumed density approximations, their use is well-justified in applications where sampling trajectories is the purpose of the application.

Codes for the methods and experiments in this paper are available at http://github.com/AaltoML/scalable-inference-in-SDEs.

## Acknowledgments and Disclosure of Funding

Authors acknowledge funding from Academy of Finland (grant numbers 324345 and 339730). We also wish to thank the anonymous reviewers for their comments on our manuscript, and Çağatay Yıldız and Xuechen Li for providing useful details on the MOCAP experiment. We acknowledge the computational resources provided by the Aalto Science-IT project. ET has been employed part-time at Sellforte Oy during the project. The MOCAP data was obtained from mocap.cs.cmu.edu (created with funding from NSF EIA-0196217).

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
