# Supplementary Material:
# Scalable Inference in SDEs by Direct Matching of the Fokker–Planck–Kolmogorov Equation

This supplementary document is organized as follows. App. A provides further details and derivations to facilitate understanding of the methodology. App. B includes full details on the experiments, baseline methods, data sets, and additional results.

## A  Methodological Details

We provide details in terms of the concept of 'solution' to an SDE, how we use a finite-differences approach for solving the FPK as baseline, and comments on the existence properties of the GP-SDE model formulation.

### A.1  On the Concept of a Solution of an SDEs

As illustrated in Fig. 1 in the main paper, the concept of a 'solution' to an SDE is broader than that of an ODE. We restrict our interest to Itô type SDEs, and consider two types of solution concepts here: *(i)* Strong (path-wise) solutions, and *(ii)* weak (in-distribution) solutions.

A strong solution trajectory $\hat{\mathbf{z}}(t)$ to an SDE resembles an 'actual' (ideal, often intractable in practice) solution trajectory. For simulation methods, their order of strong convergence $\gamma$ (see, *e.g.*, p. 132 in [40]) can be characterized by looking at the expected error after $M = 1/\Delta t$ steps of length $\Delta t$, $\mathrm{E}[|\mathbf{z}(t_M) - \hat{\mathbf{z}}(t_M)|] \leq K\Delta t^\gamma$ for some constant $K$. However, it is generally non-trivial to construct high strong order solution methods to SDEs due to the requirement of solving intractable iterated Itô integrals in the Itô–Taylor series expansion. The required step size $\Delta t$ thus remains very small; for example, the Euler–Maruyama method converges with a strong order of $\gamma = 1/2$, which makes it tricky to choose a small-enough step size to ensure the trajectories to resemble an actual solution trajectory. In machine learning, we might be interested in path-wise solutions in the case of drawing an example solution trajectory from the method that follows the evolution of a particular realization of the random forces affecting the output.

However, during training and testing time, we are typically more interested in *aggregating* properties over multiple solution trajectories to either capture the *typical* behaviour of the model or quantify *uncertainties* induced by the random forces in the model. As the model is stochastic, the full solution entails a probability distribution, $p(\mathbf{z}, t)$, depending on time $t$ and covering the space $\mathbf{z}$.

The typical approach for characterizing $p(\mathbf{z}, t)$ in machine learning applications has been through simulation (sampling), where the Euler–Maruyama scheme, the Milstein scheme, or some more general stochastic Runge–Kutta scheme is often used. For simulation methods, the weak order of convergence (see, *e.g.*, p. 137 in [40]) can be used for characterizing the method, where it is defined to be the largest exponent $\alpha$ such that $|\mathrm{E}[g(\mathbf{z}(t_M))] - \mathrm{E}[g(\hat{\mathbf{z}}(t_M))]| \leq K\Delta t^\alpha$ for any polynomial function $g(\cdot)$. This is a much weaker criterion as it only considers the error in the expectation, and for example, the Euler–Maruyama method converges with weak order convergence $\alpha = 1$. In practice, this means that the moment properties can be captured with a more moderate step-size than the path-wise resemblance of the solutions.

However, if one is only interested in the first moments and/or if the dimensionality of $\mathbf{z}$ is high, it makes sense to consider an even weaker solution concept, where one is only concerned with the first two moments of $p(\mathbf{z}, t) \approx \mathrm{N}(\mathbf{m}(t), \mathbf{P}(t))$. This is what is done in this paper.

In short, capturing the true pathwise behaviour of an SDE is challenging (NB: simulation schemes do not generally capture this well) and sampling schemes instead generally capture the distribution by a finite set of samples. If you are interested in the first moments only, it can be generally safer to model those directly.

## A.2 Approximating the FPK Solution Through Finite-differences

As a baseline in Fig. 4 we seek to seek direct ways of assessing the behaviour of the solution to the Fokker–Planck–Kolmogorov PDE or its transition density. For this we could use any tools from the vast literature of partial differential equation approximations. The approach presented here essentially uses finite differences in the input domain of $\mathbf{z}$ and then solves the resulting homogeneous ODE system directly.

The Fokker–Planck–Kolmorogov equation has the form

$$\frac{\partial p(\mathbf{z}, t)}{\partial t} = \mathcal{A}^* p(\mathbf{z}, t), \tag{18}$$

where $\mathcal{A}^*$ is the operator defined in Eq. (7) in the main paper. We can approximate this equation as a finite-dimensional system, which is a homogeneous linear system. We discretize the state space to a finite grid $\{(z_1^{(i)}, z_2^{(j)}) : i, j = 1, 2, \ldots, N\}$ and then approximate the derivatives as finite differences. The approximations can, for example, be given by

$$
\begin{aligned}
\frac{\partial p(\mathbf{z}, t)}{\partial z_1} &\approx \frac{p(z_1 + \Delta z_1, z_2, t) - p(z_1 - \Delta z_1, z_2, t)}{2\Delta z_1}, \\
\frac{\partial^2 p(\mathbf{z}, t)}{\partial z_1^2} &\approx \frac{p(z_1 + \Delta z_1, z_2, t) - 2p(\mathbf{z}, t) + p(z_1 - \Delta z_1, z_2, t)}{\Delta z_1^2},
\end{aligned}
\tag{19}
$$

and analogously in the other dimension. We can now interpret Eq. (7) through these finite difference approximations and form a (very) sparse matrix corresponding to the adjoint operator $\mathcal{A}^*$, where also the drift and diffusion terms are evaluated at the discrete values. Thus the FPK can be rewritten as a linear ODE system

$$\frac{\mathrm{d}\mathbf{p}}{\mathrm{d}t} = \mathbf{A}\,\mathbf{p}, \tag{20}$$

where $\mathbf{A}$ is the finite-difference approximation matrix for the operator $\mathcal{A}^*$. The initial conditions $p(\mathbf{z}, t_0)$ can also be collected into a vector $\mathbf{p}(t_0)$. Because our GP-SDE model is time-invariant, the solution to the homogeneous ODE initial value problem is directly given (in closed-form) as

$$\mathbf{p}(t) = \exp((t - t_0)\,\mathbf{A})\,\mathbf{p}(t_0). \tag{21}$$

To better explain how this works in practice we have added a Jupyter notebook to this supplement which reproduces this approach.

## A.3 On the GP-SDE Model Construction

The model which we call a 'GP-SDE' model in the main paper has appeared in various forms in literature before. It directly resembles a 'random' ODE model, where the random field $\mathbf{v}_\theta(\cdot)$ has previously typically either been characterized by a Gaussian random field or Gaussian process model (see, *e.g.*, [38, 13]) or some parametric model (*e.g.*, [26]). Yet, the existence of the corresponding SDE model as in Eqs. (4) and (5) is non-trivial.

The GP-SDE model is presented informally in the main paper, and we *do not* guarantee the existence of strong unique solutions to the corresponding SDE model. However, under GP increments, the weak solution of this model exists—which can also be directly empirically shown, *e.g.*, by sampling the GP in an Euler fashion vs. running Euler–Maruyama on the corresponding SDE. This highlights the *practical* aspects of this model, which is probably also why it has been appearing in various previous forms in machine learning literature.

# B  Experiment Details

We provide additional details and results for the experiments presented in the paper, and further evaluate the computational costs of linearized approximation. Fig. 10 follows the same structure as Fig. 6 in the main paper, just providing further examples from the test set.

## B.1 Empirical Wall-Clock Timing Experiments

For the timing experiments in Sec. 3, we constructed a setup that allowed us to control the approximation error. We used the Beneš SDEs model (see details on this, *e.g.*, in [40]) that has the form

$$\mathrm{d}z(t) = \tanh(z(t))\,\mathrm{d}t + \mathrm{d}\beta(t), \tag{22}$$

where $\beta(t)$ is standard Brownian motion and the initial state $z_0$ is known. This model is non-linear in the drift and the solution is not directly apparent. Conveniently, this model has a closed-form solution that we can leverage as a control. The transition density or solution to the FPK equation is given as:

$$p(z,t) = \frac{1}{\sqrt{2\pi t}} \frac{\cosh(z)}{\cosh(z_0)} \exp\left(-\frac{1}{2}t\right) \exp\left(-\frac{1}{2t}(z-z_0)^2\right). \tag{23}$$

This solution is bi-modal and thus the moment matching approach will be an approximation to the true solution distribution. Also the first two moments are available in closed-form and given by:

$$\begin{aligned} m(t) &= z_0 + \tanh(z_0)\,t, \\ P(t) &= z_0^2 + 2z_0 \tanh(z_0)\,t + t + t^2 - [m(t)]^2. \end{aligned} \tag{24}$$

This model is one-dimensional in $z$, but we expand it to $\mathbf{z} \in \mathbb{R}^d$ by considering $d$ independent Beneš SDE models over $\mathbf{z}$ with different $z_0^{(d)}$. The initial points of the trajectories were chosen with linear spacing in $[0,1]$, with a step size of $1/d$. This test setup should be favourable to a stochastic Runge–Kutta approach, where the samples now do not need to account for correlation in the latent space, thus pushing down the required number of sample trajectories, which we expect to be linear in $d$ (the assumption of a diagonal diffusion was encoded in the experiment setup *a priori*, and the independence of the dimensions in the diffusion function was not). We thus claim that this experiment rather highlights the worst case benefits of our method, rather than the best case.

The number of trajectories used in the stochastic Euler–Mauryama was chosen to match the KL divergence of the moment matching approximation. That is, we initially completed the moment matching approximation for a given dimensionality $d$, obtaining the moments $\mathbf{m}_{\mathrm{mm}}(t)$, $\mathbf{P}_{\mathrm{mm}}(t)$, which were compared to the closed-form moments in Eq. (24) by the Kullback–Leibler distance. We evaluated the KL divergence at one hundred values of $t$ (with a spacing of $0.1$), and calculated the total divergence as a sum of the divergence at individual points.

In order to determine the number of trajectories generating moments of a comparable quality to the moment matching approach, we computed the first two moments over a varying number of trajectories. The moments were compared to the true Beneš SDE moments with the same KL divergence metric as the moment approximation. For each $d$, we matched the number of trajectories $n$ by controlling that

$$\mathrm{D}_{\mathrm{KL}}\left[\mathrm{N}(\mathbf{m}_{\mathrm{EM}}^{(n)}(t), \mathbf{P}_{\mathrm{EM}}^{(n)}(t)) \,\|\, \mathrm{N}(\mathbf{m}(t), \mathbf{P}(t))\right] \leq \mathrm{D}_{\mathrm{KL}}\left[\mathrm{N}(\mathbf{m}_{\mathrm{mm}}(t), \mathbf{P}_{\mathrm{mm}}(t)) \,\|\, \mathrm{N}(\mathbf{m}(t), \mathbf{P}(t))\right], \tag{25}$$

where $(\mathbf{m}(t), \mathbf{P}(t))$ are the exact moments, $(\mathbf{m}_{\mathrm{EM}}^{(n)}(t), \mathbf{P}_{\mathrm{EM}}^{(n)}(t))$ the moments from the Euler–Maruyama solution with $n$ trajectories, and $(\mathbf{m}_{\mathrm{mm}}(t), \mathbf{P}_{\mathrm{mm}}(t))$ the moments from our moment matching approach. As sampling methods are stochastic, we generated $n$ trajectories 10 times, to account for the uncertainty. For low values of $d$, the standard deviation of the KL divergence was in the range of 10–20% of the mean value, whereas for higher values of $d$, such as 200 or 500, the uncertainty of the metric was reduced to at most a percentage of the mean.

For this experiment, where the dimensions are uncorrelated, we found that the required number of trajectories to obtain the KL divergence of moment matching approximations was, depending on the dimensionality, between $1d$ and $2d$ (for dimensions [10, 50, 200, 300, 400, 500], the number of trajectories in the same order was [25, 65, 225, 325, 440, 550]). While the linearized approximation generally was less accurate than moment matching, it was equivalent to a nearly as high number of trajectories as moment matching (for example, when $d = 200$, linearized approximation was comparable to 210 trajectories, and moment matching to 225 trajectories).

We implement the models in PyTorch [33] and report means of 10 repetitions (std over runs negligible and omitted for clarity of presentation). The GPU/CPU wall-clock times that are reported in the main paper were run on a cluster with separate GPU and CPU partitions (GPU: NVIDIA Tesla V100 32 GB with Intel Xeon Gold 6134 3.2 GHz; CPU: Xeon Gold 6248 2.50 GHz).

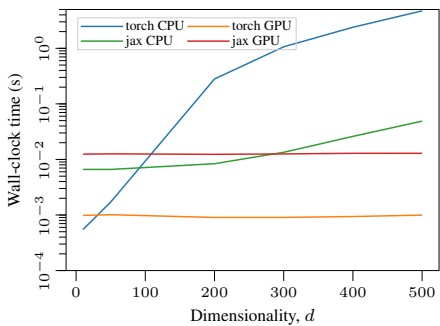

Figure 7: Empirical timing experiments of Jacobian evaluations over a single point. The variation between repetitions was negligible, and is omitted for clarity.

## B.2    Additional Jacobian Computation Timing Experiment

As we note in Sec. 2.6, a single step of linearized approximation requires only $\mathcal{O}(1)$ drift, diffusion and Jacobian evaluations. When the drift function is defined by a neural network, the scaling of the computational costs from evaluating the Jacobian is not inherently clear, as the network size is grown. In order to better assess the empirical computational costs of linearized approximation in dimension $d$, we evaluate the Jacobian of a neural network with two hidden layers, each with $3d$ nodes. The experiment results presented in Fig. 7 demonstrate that when GPU resources are available, the cost of evaluating the Jacobian in linearized approximation is not a bottleneck for growing the network size or approximating SDEs in high-dimensional spaces. The experiment set-up in terms of hardware used is as in App. B.1, and the Jacobian was evaluated 10 times, the first of which was discarded due to initialization overhead.

## B.3    GP-SDE Model Specification

This example was included to highlight the idea behind the GP-SDE model in a simple task, where encoding prior knowledge plays a major role in the outcome. We considered a GP-SDE model with just 8 observations of the dynamics, where the lack of data can be compensated for by encoding prior knowledge into the model. We use the model formulation given in Sec. 2.1, and study the effect of GP priors, the first of which is an independent squared exponential (RBF) prior for each dimension which encodes continuity and smoothness in the velocity field. The second GP prior is the multi-dimensional curl-free kernel [47]:

$$\boldsymbol{\kappa}_{\text{cf}}(\mathbf{z}, \mathbf{z}') = \frac{\sigma^2}{\ell^2} e^{-\frac{\|\mathbf{z}-\mathbf{z}'\|^2}{2\ell^2}} \left[ \mathbf{I} - \left( \frac{\mathbf{z}-\mathbf{z}'}{\ell} \right) \left( \frac{\mathbf{z}-\mathbf{z}'}{\ell} \right)^\top \right], \tag{26}$$

which encodes the assumption of a curl-free random vector field. This property can be interpreted as 'loop aversion' in the GP-SDE context. The third prior, is a multi-dimensional divergence-free kernel [47]:

$$\boldsymbol{\kappa}_{\text{df}}(\mathbf{z}, \mathbf{z}') = \frac{\sigma^2}{\ell^2} e^{-\frac{\|\mathbf{z}-\mathbf{z}'\|^2}{2\ell^2}} \left[ \left( \frac{\mathbf{z}-\mathbf{z}'}{\ell} \right) \left( \frac{\mathbf{z}-\mathbf{z}'}{\ell} \right)^\top + \left( (d-1) - \frac{\|\mathbf{z}-\mathbf{z}'\|^2}{\ell^2} \right) \mathbf{I} \right], \tag{27}$$

which encodes the assumption of no divergence in the random vector field. This property can be interpreted as 'energy preservation' or source-freeness. These properties are visible in Fig. 3, where the hyperparameters are fixed to same values for all models (even if the interpretation differs).

## B.4    Synthetic Race Track

As an additional example, we run the proposed algorithm on two synthetic race-tracks: oval-shaped and bean-shaped. For both the experiments, we have a true race track and a set of noisy observed race tracks. The latent state, $\mathbf{z} \in \mathbb{R}^2$, governs the dynamics in the original space. Further, using the noisy observed race tracks, we create a set of observation vectors on which the GP is conditioned. For this, we use the Gaussian process regression model in the GPflow [28] framework with the

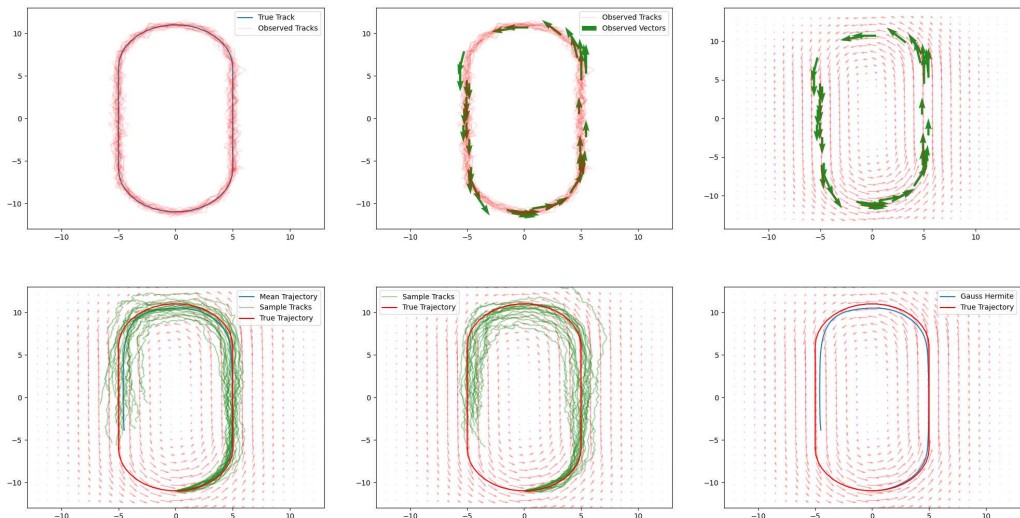

Figure 8: The output of oval shaped synthetic car race track. (a) True track and noisy observation tracks. (b) Observation vectors and observation tracks. (c) Mean predicted vector field of the GP-SDE. (d) Trajectories predicted using the Euler scheme for the 'random ODE' interpretation, with dynamics drawn from GP samples. (e) Trajectories predicted by the Euler–Maruyama scheme for the GP-SDE model. (f) Mean trajectory path predicted using Gauss–Hermite quadrature by assumed density.

squared-exponential (RBF) kernel. The hyperparameters of the model are optimized with the Adam optimizer with a learning rate of $0.001$, and the kernel hyperparameters are initialized with default values, length-scale and variance $1.0$.

Fig. 8 and Fig. 9 showcase the outputs on the two synthetic race tracks. The figures show the true track and noisy observations of the dynamics along the trajectory. On top-right, the mean predicted vector field of the GP-SDE for these observations. Then we visually compare trajectories predicted using the Euler scheme for the 'random ODE' interpretation, where the dynamics are drawn from GP samples (see Eq. (1) in the main paper), with trajectories predicted by the Euler–Maruyama scheme for the GP-SDE model. Finally, we plot the mean trajectory path predicted using $4^{\text{th}}$ order Gauss–Hermite quadrature by an assumed density assumption.

## B.5   Rotating MNIST

In the spirit of Fig. 2, we run the proposed methods on Rotating MNIST ([25], available under CC BY-SA 3.0), similar to [49, 5]. The data set consists of various handwritten digit '3's rotated uniformly in 64 angles. The training data set is generated by randomly selecting 180 different versions of digit '3' resulting in the total size of the training data set to be $11,520$ images. A separate set of 20 digits are chosen to form a test set. We train a VAE [21] first by freezing the latent space dynamics, and then freezing the VAE encoder/decoder and training a 16-dimensional GP-SDE model in the latent space with independent squared exponential GP priors (see App. B.5). We implement the VAE model in PyTorch [33] and use GPflow [28] for the latent GP model. The latent space is chosen to be $d = 16$ dimensional and a sparse variational Gaussian process (SVGP) model [15] with 1000 trainable inducing points is leveraged to scale the GP training. We use independent squared-exponential prior covariance function for each latent dimension dynamics. The models are trained with the Adam optimizer [20] (learning rate $0.001$), and the VAE loss function is a weighted sum of binary cross-entropy and KL-divergence, whereas the GP objective function is the ELBO. The training of the two components is disjoint, and the GP is trained on a fixed latent space given by a trained VAE.

An output of a test point is illustrated in Fig. 6, where we feed in one observation and let it follow the learned dynamics of rotation. As the baseline, we use 1000 trajectories computed using Euler–Maruyama with step length $0.1$. Fig. 6a demonstrates the model's capability to learn the latent trajectory, and we show the trajectories for all the methods in three latent dimensions with the

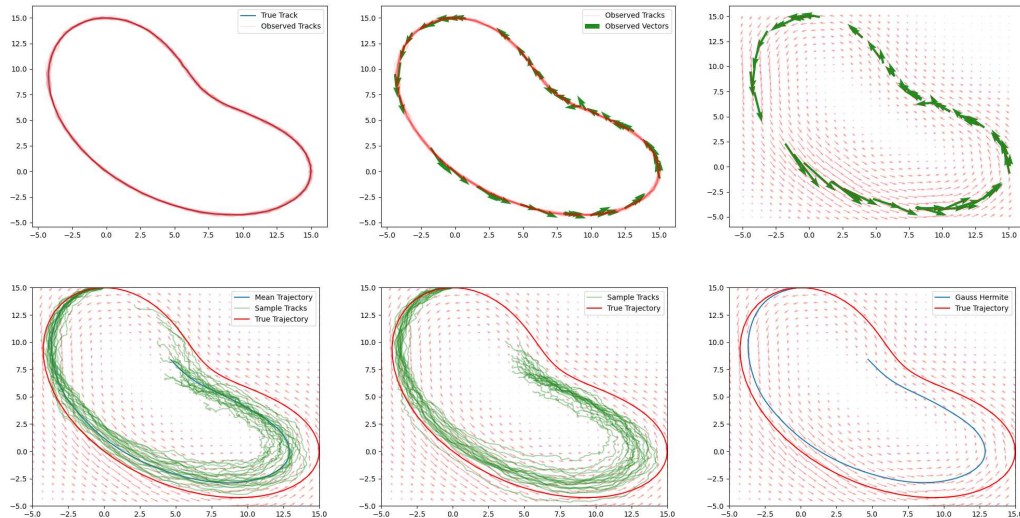

Figure 9: The output of bean shaped synthetic car race track. (a) True track and noisy observation tracks. (b) Observation vectors and observation tracks. (c) Mean predicted vector field of the GP-SDE. (d) Trajectories predicted using the Euler scheme for the 'random ODE' interpretation, with dynamics drawn from GP samples. (e) Trajectories predicted by the Euler–Maruyama scheme for the GP-SDE model. (f) Mean trajectory path predicted using Gauss–Hermite quadrature by assumed density.

most variation. The trajectories for the three methods overlap exactly, which also shows in Fig. 6b that shows the generated outputs in the observation space together with the associated marginal uncertainties. We include further details in Fig. 10.

For quantitative comparison, we do a 10-fold cross-validation study on the rotating MNIST data. The full dataset consists of 200 randomly chosen digits '3' which are split into 10 folds, each fold consisting of 180 training and 20 test digits. As discussed, each digit is uniformly rotated around 64 angles thus making the training dataset size equal to 11,520. Both the models, VAE and latent-GP, are trained independently with the same initial hyperparameter values and an equal number of epochs over different folds. On the test dataset, we perform inference via three schemes: Euler–Maruyama, linearization, and moment matching. Euler–Maruyama acts as a baseline for which 1000 samples with 0.1 step-size are generated. For linearization and moment-matching we use the Euler scheme with 0.1 step-size for solving the resulting ODE. The MSE values are calculated for the mean. Alongside mean we also characterize the uncertainty estimates by studying the negative log probability density (NLPD) for all the three schemes in the latent space over time. The MSE and (final-step, $t = 64$) NLPD results are in Table 1 in the main paper, where we see that the sampling scheme performs slightly better, especially in terms of NLPD—even if the qualitative results did not show a clear difference.

### B.6 Motion Capture Experiment Details

For the motion capture experiments, we used the same pre-processed CMU Walking data set as in Yıldız et al. [49]. The relevant hyperparameters and design choices were the weighting of KL-divergence, learning rate of the optimizer, neural network designs, choice of SDE approximation, choice of the ODE solver, and choice of the prior process for regularization.

For weighting the KL-divergence, we tested the values $\gamma = \{1.0, 0.1\}$, both with a linear schedule until epoch 200 and fixed value. The listed MSE was achieved when using $\gamma = 1$ fixed and with a learning rate of 0.01. The drift, diffusion, encoder and decoder networks were all trained simultaneously in 1500 iterations using the Adam optimizer, see Fig. 11 for the detailed network designs. With the exception that we model the change in the latent context, the neural networks are similar to those presented in Li et al. [26].

For the prior process, we used a simple stochastic process with zero drift and $0.1\mathbf{I}$ diffusion. In our experiments, we found that using a prior process is fundamental for successful training over an SDE

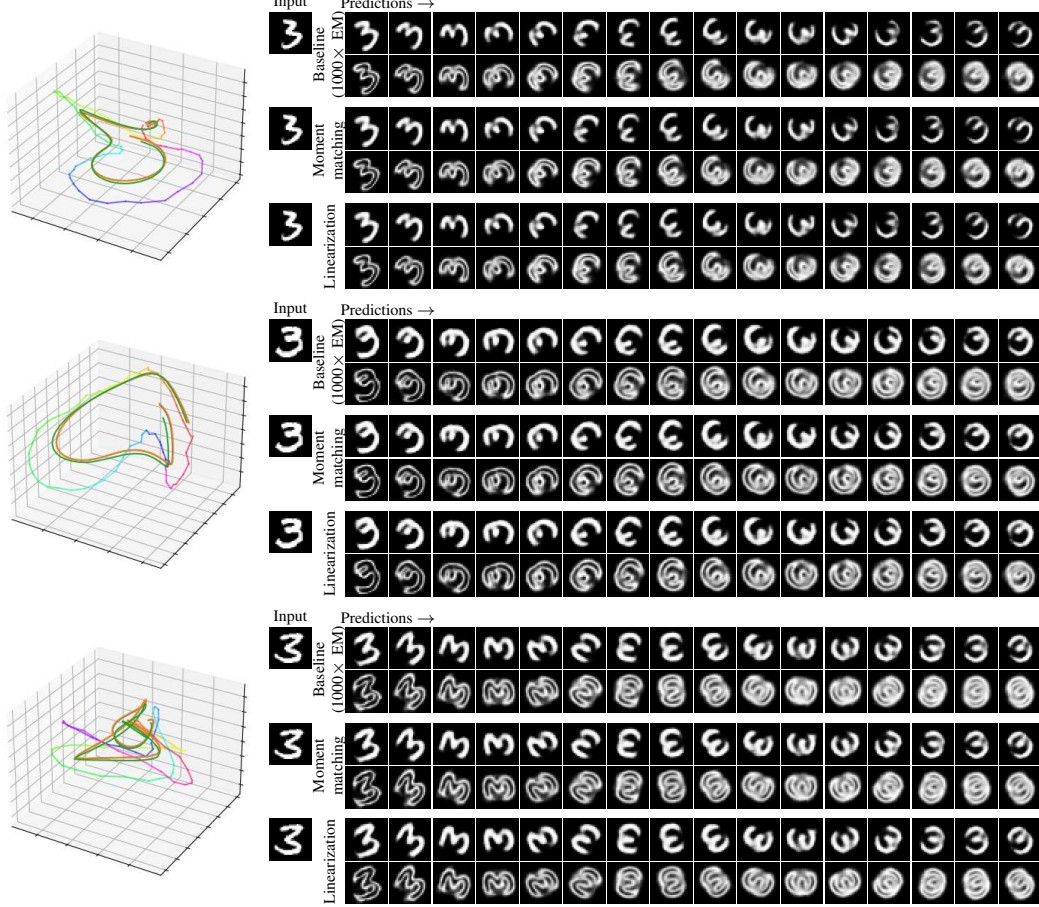

Figure 10: Further test set example results on the rotating MNIST data set. The left-hand side shows the prediction trajectories for one test set image in the latent space. Evolution of the true trajectory is shown in HSV colour code and the predicted trajectory by Euler–Maruyama, moment matching, and linearization scheme is shown in green, blue, and orange, respectively (all overlapping one another). The right-hand figures show the progression of the prediction (mean and marginal std images) of the test set image input, when it traverses the learned dynamics in the latent space. Both the moment matching and linearization schemes match the baseline (computed with 1000 Euler–Maruyama trajectories).

approximation: optimizing solely for maximum likelihood resulted in unrealistic parameter values and lead to numerical instability. As an alternative to zero drift prior processes, we tested a trainable drift network, and inspired by Li et al. [26], a prior diffusion network that matches the posterior in state dimensions. While the alternative prior processes produced a more informative latent space, we chose the zero drift prior process both for its simplicity and performance: the lowest MSE was achieved when using a zero drift prior. The selected ODE solver was a $5^{\text{th}}$ order Runge–Kutta method. When running the model implementation for a linearized approximation on TensorFlow and a Tesla V100 GPU, training was completed in approximately 2 hours and 45 minutes.

We also include a separate timing comparison between the methods under this model, where we control for equal step size and between methods and using the plan Euler/Euler–Maruyama scheme. The results in Table 2 show timings for one pass with a PyTorch implementation and using the same hardware as presented in App. B.1.

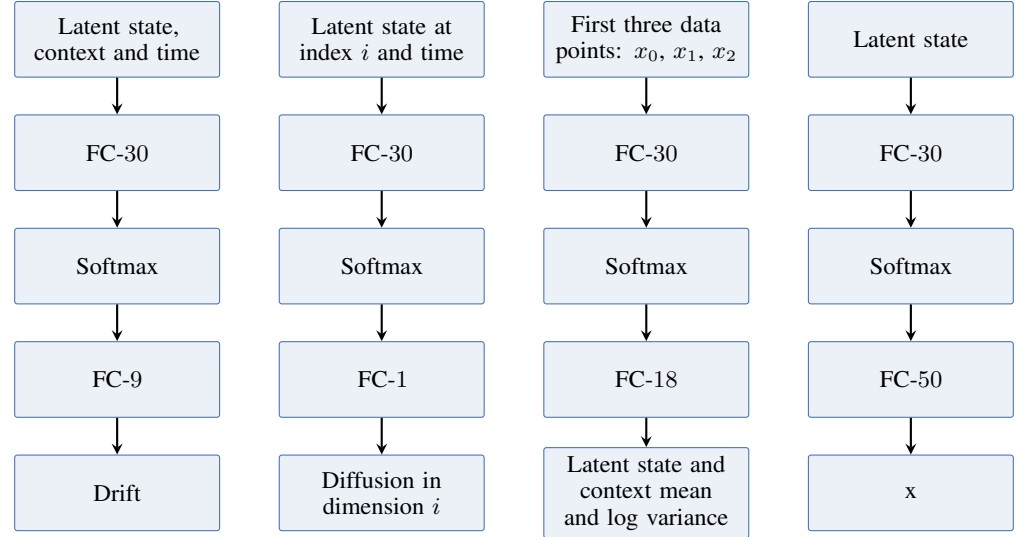

Figure 11: Neural network designs (from left to right) for the drift, diffusion, encoder and decoder networks. The diffusion network is duplicated 9 times, one for each latent state or context dimension.

## C   Author Contributions

The original idea and motivation for this work was conceived by AS, who also wrote a first draft of the paper. ET had the main responsibility of writing the related work section, the computational complexity section, and the MOCAP experiment. PV worked on the GP-SDE models and had the main responsibility of the rotating MNIST experiment. All authors contributed to finalizing the manuscript.