# OpenReview forum: "Scalable Inference in SDEs by Direct Matching of the Fokker–Planck–Kolmogorov Equation"
_NeurIPS.cc/2021/Conference — NeurIPS 2021 Poster_

### Official Review · Reviewer_qke7 · 2021-07-13

**Rating:** 8
**Confidence:** 4

**Summary:**

In this paper, a fast, scalable inference method in stochastic differential equations is introduced, which proceeds by approximating the solutions of the Fokker-Planck-Kolmogorov (FPK) equation instead of relying on sampling schemes. In particular, the solutions to the FPK system is approximated by a Gaussian, whose mean and covariance are shown to satisfy a system of ODEs, which are then further approximated by linearisation or moment matching. The authors demonstrate that solving these approximate systems have significant computational advantage to sampling directly from the SDEs, especially in high dimensional settings and yield qualitatively similar results on experiments on the rotating MNIST dataset and motion capture data.

**Limitations And Societal Impact:**

I do not see any potential negative use of this work. In terms of limitations, the quantitative results of the proposed method comes at a cost of the gain in computational speed as shown in the experiments in §4. However, the baseline performance is expected to decrease with fewer sample size so it might have been interesting to investigate the smallest number of samples that are required in the baseline before it outperforms moment-matching/linearisation.

**Main Review:**

Originality:
While sharing similarities with [26], I believe that this paper manages to develop an original approach to learning SDEs by starting from the Fokker-Planck equations and deriving everything from there, which I haven't seen yet in the literature. I find the idea of combining this technique with VAEs interesting as indeed, a Gaussian approximation is necessary anyways in the encoder. The GP-SDE idea is also interesting, however I am not fully convinced that this equivalence between random ODEs and SDEs holds (more details below).

Quality:
I believe that overall the paper is relatively easy to read and is of high standard in terms of intellectual contribution and experiments. However, there were some places that were unclear to me, in addition to a few misprints as I list below.

Clarity:
Below I list some parts of the paper that were unclear to me or what I believe are misprints:
- The matrix $K_*$ is not defined in §2.1
- Similarly in §2.1. Is $Q$ the precision matrix? It seems to appear out of nowhere.
- $z_1$ in (5) should be $z_*$?
- I am unsure whether the claim about the equivalence between the random ODE with a GP vector field and SDEs in §2.1 is correct. In [13], they only show the relation between the two systems under the Euler-Maruyama discretisation, where the discretised transition probability can be shown to be the same. However, this certainly does not imply the equivalence in probability for the full continuous system, and I have not seen a rigorous treatment of the equivalence between the two systems.
- I think some experimental details for the rotating MNIST and video motion data are lacking:
  - For the rotating MNIST, it wasn't clear to me how the training took place in the latent space. From my understanding, the GP-SDE is trained using the dataset $D = (z_i, \Delta z_i)$, but how do you obtain this data in the latent space from training data in the observation space?
  - In the video motion experiment, $p$, $q$ and $\gamma$ in the loss (17) are not defined.
- Line 310. what is "hw description"?
- Figures
  - What are the red vector fields in figure 3? Is it the GP posterior mean?
  - In figure 6 a), are the plotted paths the mean trajectories of the SDEs or the sample paths (I presume the former given how smooth it is)? This was not specified in the body.
  - How many Euler-Maruyama trajectories are used in the top panel of figure 5?

Significance:
I can see that the techniques introduced in this paper may be useful in situations when a single SDE run is extremely costly. For example in weather prediction, typically only a few ensemble runs are feasible, yet having good uncertainty quantification is vital so there may be some potential use case in this area.


**Time Spent Reviewing:**

6

---

> ### Author Response · Authors · 2021-08-09
> **Response to qke7**
>
> We thank the reviewer for their enthusiasm and comments on the manuscripts and the detailed list of possible clarity improvements. We intend to improve the clarity of the text with the additional space provided for the camera-ready version, should the paper be accepted. We have fixed the typos pointed out, and will address the remaining questions and concerns in the order received.
>
> * In Sec. 2.1, the matrix $Q$ is the spectral density of the white noise process or Brownian motion. We limit the interest here to unit Brownian motion and thus $Q = I$ as is often done in ML papers.
> * Regarding the relationship between the random ODE and the GP-SDE model: As in [13], we do not intend to make strong claims on the equivalence (we have some additional discussion on this in the appendix). The time-marginals, however, match in practice, and the analysis could most likely be extended by constructing the relationship between the two under suitable assumptions on the GP model.
> * In the rotating MNIST experiment, the encoder and decoder are first trained during a pre-training phase before the latent dynamics. The training of the dynamics thus happens in the encoded latent space. We have tried to cover the details in the appendix, but we will go through this again.
> * The unclear terms in the loss function used for the MOCAP experiment are defined as follows: $p$ is the likelihood of the true observations given latent codes from the model, $q$ is the latent prior at time 0, and $\gamma$ is a hyperparameter controlling how the regularization is weighted.
> * By “hw description”, we meant “hardware description”, which we agree is too ambiguous in the shortened form.
> * The red vector fields in Fig. 3 are indeed the GP posterior mean. We now mention this.
> * In Fig. 6a, the plotted paths are indeed the mean trajectories. We now mention this.
> * In Fig. 5 (top), the number of Euler–Maruyama trajectories was chosen so that it matches the accuracy (in terms of KL divergence) of the assumed density approximation. The number of required trajectories in the plot was between 1–2.5 times the dimension, for example for $d=10$ it was 25, whereas for $d=200$ there were 225 trajectories. NB: This is exactly what is also proposed in the review (i.e., “the smallest number of samples that are required in the baseline before it outperforms moment-matching/linearisation”). See App. B.1 for a detailed explanation.

---

> > ### Comment · Reviewer_qke7 · 2021-08-30
> > **Rebuttal response**
> >
> > I thank the authors for the response. All of my questions are addressed and my outlook of the work remains positive after reading the other reviews and discussions. I maintain my score of 8.

---

### Official Review · Reviewer_P9hS · 2021-07-14

**Rating:** 6
**Confidence:** 4

**Summary:**

The paper studies the GP-SDE model, and proposes to approximate the marginals with Gaussians whose mean and covariance are approximations of the true quantities.

**Ethical Concerns:**

This is a strict methodology paper. This discussion does not apply.

**Limitations And Societal Impact:**

This is a strict methodology paper. This discussion does not apply.

**Main Review:**

The paper studies ways of Gaussian assumed density approximations can be made practical for dynamics models with an SDE component. In this regard, the paper is novel.

The paper is of fair quality, is written reasonably clearly, and likely would have some influence in this area. However, the paper does possess certain drawbacks, mostly in terms of technical details.

Additional questions/comments:
- The motion capture data experiment somewhat seems detached from most of the paper, since the model in the latent space here isn't a GP anymore. Moreover, the objective (17) isn't that of that of the latent SDE model either, since the latent space penalty is only applied at timepoints with observations, as opposed to the entire time horizon of consideration.
- line 163-166: Could you clarify why the choice is contradictory? The encoder of the latent SDE model outputs a drift function given the observations; this is not a variational Gaussian approximation.
- Section 2.6: I appreciate there being a section dedicated to computational cost.

  a. However, (line 217) the time to evaluation the drift and diffusion is likely quite different from the time needed to evaluate the Jacobian, so I wouldn't put them together unless there's a good reason.

  b. (Line 223): I don't quite get why the sample complexity is exponential in d. This issue seems more nuanced as presented, and likely depends on the divergence/metric considered. It seems the local squared error should scale polynomially in d, in which case the strong and weak errors might just not be exponential. If one is only concerned with approximating functionals, then Monte Carlo sampling shouldn't be terrible.

**Time Spent Reviewing:**

3

---

> ### Author Response · Authors · 2021-08-09
> **Response to P9hS**
>
> We thank the reviewer for their comments on the manuscript. We appreciate the detailed feedback and requests for clarifications on the computational experiments and intend to use the ninth content page of the camera-ready version for improving the clarity of the text, if the paper is accepted. We address the questions and concerns in the order received.
>
> * The purpose of the MOCAP experiment is to compare our approximation methods to earlier works in ODEs/SDEs, where neural networks have been favoured over GPs. Therefore, we chose to use an SDE model with neural networks as drift and diffusion for a more direct comparison, with the additional goal to highlight that while we focus on GP-SDEs for most of the manuscript, the approximation methods presented are general and the computational benefits apply more widely.
> * On lines 163–166 the choice of wording could be better. We mean to refer more widely to the fact that it might be unnecessary to sample realizations, if the interest is only in the time-marginals of the process.
> * It is true that evaluation of the function itself and the Jacobian is more costly than just evaluating the function. This fact is also hidden away in the constant in the big-O notation. However, the cost of evaluating the Jacobian is marginal in practice as also shown in the empirical timing results for the linearised approximation. We will nevertheless add a comment regarding this in the paper.
> * It is true that the exponential scaling in $d$ is actually more nuanced, and we are happy to modify the wording. In any practical application, the scaling should certainly be better than exponential, and thus we have tried to make an honest example by choosing a model for the example comparison (Sec. 2.6 / Fig. 5), where the dimensions are independent and the problem being an easy one for the sampling schemes.

---

> > ### Comment · Reviewer_P9hS · 2021-08-28
> > **update**
> >
> > I read other reviewers' comments and authors response. I thank the authors for trying to address my comments.
> >
> > My main concern regarding the Jacobian evaluation remains. The way vanilla autodiff is designed makes instantiating the Jacobian d times as costly as just computing a vector-Jacobian product, and rougly 2d-3d times as costly as computing the function it self. See some introductory tutorials on autodiff for some more precise estimates.
> >
> > Obviously, there are ways of obtaining the Jacobian much faster in practice, but it either takes in practice time O(d) or memory O(d) assuming the time cost of vJp and function evaluation is O(1).
> >
> > > However, the cost of evaluating the Jacobian is marginal in practice as also shown in the empirical timing results for the linearised approximation.
> >
> > So I would say this essentially sweeps the major issue under the rug and could be potentially misleading. How does this Jacobian cost scale as you make f as a neural net larger? How does this cost scale as you increase the dimensionality of the input/output space? What is being implemented in practice?

---

> > > ### Author Response · Authors · 2021-08-29
> > > **Re: update**
> > >
> > > Thank you for getting back to us. In retrospect, our initial reply was not detailed enough, and we try to clarify the remaining concerns below.
> > >
> > > > The effect of Jacobian evaluation of the linearization approach to computational scaling
> > >
> > > How the computational scaling is currently presented at the beginning of Sec. 2.6, is in terms of the number of required parallel executions of the model, which indeed is $O(1)$ evaluations of the drift, diffusion, and Jacobian per step for the linearization approach. We agree that this feels backwards, and the right way would be to also cover the cost of *actually evaluating* the Jacobian as well. As promised in our earlier reply and highlighted in your update, we will revise this part to show the explicit dependence on $d$ in the linearization approach.
> > >
> > > > Practical meaning implementation-wise
> > >
> > > For GP-SDE models, the Jacobian is typically easy to evaluate in closed form, while for neural nets we use Jacobians through `torch.autograd` or TensorFlow. In the empirical studies, the linearization approach is less accurate, but faster compared to sampling or matching the moments. This is what we meant by the effect of evaluating the Jacobian to only be *marginal* (still order of magnitude faster than calculating a Cholesky decomposition on every step as required for the moment matching).
> > >
> > > The timing experiment in Fig. 5 already considers varying the dimensionality, but we see your point in doing the same for a large neural network model with strongly dependent dimensions and a more complicated structure. We would be happy to include such an experiment in the appendix.

---

> > > > ### Comment · Reviewer_P9hS · 2021-08-29
> > > > **update v2**
> > > >
> > > > Thank you for getting back.
> > > >
> > > > My specific point about the Jacobian was in particular related to when using standard autodiff primitives like `torch.autograd.grad` or `tf.gradients`.
> > > >
> > > > These frameworks are primarily based on reverse-mode autodiff, and hence instantiating the entire Jacobian requires O(d) vector-Jacobian products. The same complexity scaling is true for forward-mode autodiff as well, assuming the input and output dimensions are the same.
> > > >
> > > > There are some more updated versions of these primitives in libraries like JAX, where O(1) call to the gradient function might give you the Jacobian, but that's at the cost of increased memory usage -- in some sense, this is now O(d) time as costly as a function evaluation in terms of memory.
> > > >
> > > > In either case, I don't quite think you can group the number of function evaluations and the number of Jacobian evaluations together, since these have fundamentally different scalings wrt the dimension.

---

### Official Review · Reviewer_vsyv · 2021-07-16

**Rating:** 6
**Confidence:** 3

**Summary:**

The paper presents a practical methodology for SDE-based inference in machine learning that avoids the use of Monte-Carlo sampling. The methodology invokes a Gaussian density assumption (often used in latent space modelling) to directly approximate the first two moments of the SDE solution. This is achieved by solving a system of ODEs obtained from the Fokker-Planck-Kolmogorov equation. Empirical evidence shows that this scales well with dimension and the approach is demonstrated on Rotating MNIST and motion capture datasets. The motion capture experiment shows the proposed approach achieves competitive accuracy to previous ODE and sampling-based SDE methods, whilst being substantially faster than the latter.

**Limitations And Societal Impact:**

The authors address that their approach is not suitable for problems where pathwise sample trajectories are required. I think they could give an example, such as using SDEs for generative time series modelling (Kidger et al., (ICML 2021)). Whilst the authors argue that a Gaussian density assumption makes sense for latent space models, they do not address its potential limitations.

**Main Review:**

Strengths:

* The proposed methodology is an interesting and appealing alternative to sampling-based methods for SDE inference.

* Experiments on synthetic and real-world datasets show the assumed density approach scales well with dimension and has competitive accuracy compared to other ODE and SDE models. In particular, the proposed methods are substantially faster than sampling-based SDE models.

* The paper is clearly written.

Weaknesses:

* I am unsure about the originality of the work. As noted by the authors, “The model which we call a ‘GP-SDE’ model in the main paper has appeared in various forms in literature before” and the methods in Section 2 can all be found in Section 9.1 of (Särkkä and Solin, 2019). In the machine learning setting, the linearizing and moment matching via cubature approaches are presented and empirically tested in Sections 4 and 5 of [Look et al. (2020)](https://arxiv.org/pdf/2006.08973v1.pdf). However, recent versions of (Look et al.) no longer discuss these approaches.

* [Look et al. (2020)](https://arxiv.org/pdf/2006.08973v1.pdf) claim that discretizing the SDE before moment matching has advantages over matching moments of the Fokker–Planck–Kolmogorov equation. They argue this with a stability result as well as empirical evidence. A similar discussion would be helpful here.

* In the motion capture experiment, the latent SDE drift and diffusion are taken to be neural networks instead of the mean and covariance functions of a Gaussian process. As a result, I believe the paper should be written with more generality. The key idea is that an assumed density approach can be advantageous over sampling, and this is compatible with both GPs and neural networks.

* The authors claim that the proposed method approximates the mean and covariance of an SDE solution. I believe this claim is entirely heuristic. In particular, there are no theoretical guarantees about the accuracy of the assumed density approximations.

Minor comments:

* In the abstract and introduction, the authors claim that sampling can suffer from the "Curse of Dimensionality" (CoD). In the many areas of numerical analysis, the opposite is true and sampling breaks the CoD. For example, the paper "Solving high-dimensional partial differential equations using deep learning", Han et al. (2018) uses SDE sampling to break CoD when solving PDEs. I think it would be more mathematically accurate just to say that sampling can be inefficient in high dimensions (i.e. leave out CoD).

*  It is not explained where the assumed density approximation (equations (9) and (10)) comes from, though the subsection titles imply that it is the Fokker-Planck-Kolmogorov equation. A brief explanation of this would be helpful.

* The authors do not explicitly define $\mathbf{K}_\ast$ in Section 2.1.

Conclusion:

The paper presents an attractive alternative to the standard Neural SDE methodologies as it avoids costly Monte Carlo simulation. Experimental evidence suggests the proposed methods scale well with dimension and have comparable accuracy to alternative ODE and sampling-based SDE methods (whilst being significantly faster than the latter). Thus, I believe the underlying methodology has great potential. On the other hand, I think the paper should present its methodology in more generality (i.e. discuss Neural SDEs as well as GP-SDEs) and better address its similarities with [Look et al. (2020)](https://arxiv.org/pdf/2006.08973v1.pdf).

Typos:

* ".To" (page 4)
* "see hw description" (page 9)

**Time Spent Reviewing:**

3

---

> ### Author Response · Authors · 2021-08-09
> **Response to vsyv**
>
> We thank the reviewer for their detailed review and good comments on the manuscript. We will address the concerns and comments in the order received.
>
> * We agree with the comment that Gaussian approximations are widely-applied in SDE literature, and Look et al. have also discussed these before in their work. We try to be very open regarding this in the paper.
> * The previous version of Look et al. (2020) (we also need to update our citation to point to this specific archived version of their paper) claimed discretization improves stability. This can, most likely, be the case in some cases, but we aimed for a *direct* approximation of the first moments of the FPK, where the model is not actually discretized even after, but the corresponding ODEs are *solved* at discrete points. We will refine the paper and add discussion regarding the relation to Look et al.
> *As stated in the review, the key idea is that an assumed density approach can be advantageous over sampling, and this is compatible with both GPs and neural networks. This is also why we included the MOCAP experiment, for completeness and demonstrating that the assumed density approach is not limited to the models with GPs. We have tried to make this explicit in the Introduction and in how we split Sec. 2.1 and 2.2. We are happy to improve the paper by using the additional space in the camera-ready submission to make this more clear.
> * We agree with the reviewer that the guarantees for the approximated Gaussian integrals (by quadrature) to capture the evolution of the true moments are almost non-existent (for example, the 3rd order cubature is exact for polynomials up to order 3, which is of limited use here for providing guarantees). However, under suitable practical conditions, this type of approximations have successfully been applied for decades in non-linear signal processing and they are known to perform rather reliably in practical applications. However, we think stating this explicitly and including some discussion on this is a very good idea.
> * We see the point regarding re-wording things regarding the CoD and will revise the paper accordingly.
> * We also thank the reviewer for pointing out other minor corrections. We will clarify these points.

---

> > ### Comment · Reviewer_vsyv · 2021-08-30
> > **Update**
> >
> > I thank the authors for their response. After reading through the manuscript and the other reviews/responses, I have fewer concerns regarding the originality of the research. Since a discussion regarding the relation to Look et al. (2020) will be added, I am leaning towards acceptance and will raise my score to 6.

---

### Official Review · Reviewer_nBa9 · 2021-07-16

**Rating:** 7
**Confidence:** 4

**Summary:**

This paper uses random ordinary differential equations (ODEs) to specify stochastic differential equations (SDEs), where Gaussian processes (GPs) are used to define the vector fields. It is well-known that the dynamics of probability density of states driven by SDEs is given in form of Fokker-Planck equation which is a partial differential equation. In general, there is no closed-form solution of Fokker-Planck equation and the solution is derived by SDE solvers such as Euler–Maruyama (EM), Milstein, and stochastic Runge–Kutta (SRK). These solvers are expensive and inefficient to use in high-dimensional settings. This paper proposes to use an approximate solution by assuming a Gaussian density and tracking the first two moments, mean and covariance matrix. Then the authors derive two ODEs defining the dynamics of mean and covariance matrix. To simplify the ODEs, the authors suggest two schemes: 1- linearizing the Fokker-Planck equation and 2- matching the moments of Fokker-Planck equation. The first scheme is a local approximation and so is less accurate than the second one. However, in terms of runtime, the linearized method is faster than the moment matching one. Overall, the proposed method in this paper can be used to 1- apply prior knowledge to the vector field (trajectories) by using different GP priors and 2- do an efficient inference of probability density function of an SDE by assumed density (Gaussian) approximation of Fokker-Planck equation and deriving two ODEs for the mean and covariance of the distribution which can be solved using ODE solvers and without any stochastic solvers like SRK.

**Limitations And Societal Impact:**

The authors should discuss the limitation of their work in the paper. Their method is good in the sense that it bypasses the use of costly stochastic SDE solvers, however, there is a trade-off between runtime and accuracy and it should be fairly discussed in the paper. I, as a reader, was not fully convinced, based on the results on MOCAP, that this methods is working better than latent ODEs.

**Main Review:**

- The paper is clear and well written. It tries to address an important problem in neural SDEs: how to efficiently learn the dynamics and avoid expensive stochastic SDE solvers. The authors use the existing methods and tricks to address this issue. They use GPs to model the SDEs as random ODEs and employ the assumed density approximation tricks to estimate the pdf of the resulting SDE which evolve based on Fokker-Planck equation. Both techniques are not novel and the authors have used already known techniques to approximate the pdf of the SDEs without stochastic solvers. Therefore, the main contribution of the paper is not method advancement, but rather using the other methods to 1- improve the efficiency of learning SDEs by removing the need of stochastic solvers, 2- eject prior on the vector field of the ODEs via modeling them as GPs. The proposed method could be a practical improvement by simplifying the solutions of SDEs.


- Some of the results do not show improvement. For example, MOCAP results show that the test MSE is more than that of latent ODE. As one of the reasons of using SDEs instead of ODEs is their ability to capture the uncertainty and overall their better performance in complicated systems (as seen by improvement of latent SDE over latent ODE), having worse performance than latent ODE questions the proposed method's ability in capturing the uncertainty accurately. I think the authors should explain why this is the case and provide more convincing examples. Otherwise, even though this method speeds up learning the SDEs, one could use ODEs if their performance is on par with this method.

**Time Spent Reviewing:**

10

---

> ### Author Response · Authors · 2021-08-09
> **Response to nBa9**
>
> We thank the reviewer for the thoughtful review and comments on the manuscript. We find the characterization of the paper to be a good summary of its key aspects.
>
> The main concern was related to the MOCAP experiment, where we did not show superiority in comparison to other methods. This is certainly true, but this experiment was included mostly for completeness and in order to show how the proposed assumed density approach works on a benchmark task beyond the GP-SDE kind. The model is, however, not exactly the same as in [25] as they have not made their implementation details available and we did not manage to replicate their MSE result. However, compared to earlier ODE models, like the ODE2VAE [47], the MSE performance improved. Still the main point, regardless of the exact model, is the gains in compute time which are illustrated in Table 2.
>
> We also acknowledge the comment regarding discussion of possible limitations, and we are happy to use the additional space in the camera-ready submission to cover discussion on trade-off between possible benefits and pitfalls of pathwise and weak solutions in different scenarios across combinations with various ML applications.

---

> > ### Comment · Reviewer_nBa9 · 2021-08-29
> > **Update**
> >
> > I appreciate the authors for their answer. My concern regarding the MOCAP experiment is addressed. I raise my score to 7 and I believe the paper is good enough to be accepted.

---

### Decision · Program_Chairs · 2021-09-27

**Decision:**

Accept (Poster)

**Comment:**

This paper is concerned with inference of neural stochastic differential equations. In general, exact inference in these is intractable. This paper proposes a pragmatic technique, whereby the discretization is modeled by a recursive (deterministic) Gaussian approximation. They show computational advantages, particularly in higher dimensions.

All reviewers felt this was a valuable method, with a worthy contribution. The main concerns that were identified were relationships to some existing work (largely addressed in feedback) some unclear notation (also addressed in feedback), and some questions about the computational cost of the Jacobian step. Here, we echo the reviewer in that it is not reasonable to treat function evaluations and Jacobian evaluations as equal.  The authors are trusted to fulfill their commitment to including a large neural-network model with strong dependencies to consider this timing issue experimentally. Still, overall the method appears correct, plausibly useful and the paper is well-written, thus I recommend acceptance.